# Flexible and cost-effective genomic surveillance of *P. falciparum* malaria with targeted nanopore sequencing

Mariateresa de Cesare [1,7], Mulenga Mwenda[2,7], Anna E. Jeffreys [1], Jacob Chirwa[3], Chris Drakeley[2], Kammerle Schneider[2], Brenda Mambwe[2], Karolina Glanz [4], Christina Ntalla[4], Manuela Carrasquilla[4], Silvia Portugal [4], Robert J. Verity [5], Jeffrey A. Bailey [6], Isaac Ghinai[1], George B. Busby [1], Busiku Hamainza[3], Moonga Hawela[3], Daniel J. Bridges [2] & Jason A. Hendry [1,4] ✉

Genomic surveillance of *Plasmodium falciparum* malaria can provide policy-relevant information about antimalarial drug resistance, diagnostic test failure, and the evolution of vaccine targets. Yet the large and low complexity genome of *P. falciparum* complicates the development of genomic methods, while resource constraints in malaria endemic regions can limit their deployment. Here, we demonstrate an approach for targeted nanopore sequencing of *P. falciparum* from dried blood spots (DBS) that enables cost-effective genomic surveillance of malaria in low-resource settings. We release software that facilitates flexible design of amplicon sequencing panels and use this software to design two target panels for *P. falciparum*. The panels generate 3–4 kbp reads for eight and sixteen targets respectively, covering key drug-resistance associated genes, diagnostic test antigens, polymorphic markers and the vaccine target *csp*. We validate our approach on mock and field samples, demonstrating robust sequencing coverage, accurate variant calls within coding sequences, the ability to explore *P. falciparum* within-sample diversity and to detect deletions underlying rapid diagnostic test failure.

The malaria parasite species *Plasmodium falciparum* is an example of both the potential value of genomic surveillance and the obstacles that can impede its implementation. Although a variety of antimalarial drugs exist, the evolution of resistance has compromised their efficacy[1,2]. Most critical is resistance to artemisinin, the dominant chemotherapeutic agent in artemisinin-based combination therapy (ACT) and the foundation of global guidelines for the treatment of malaria[3]. Formerly confined to the Greater Mekong Subregion[4–6], genetic mutations associated with artemisinin resistance have recently been detected in Uganda[7] and Rwanda[8], escalating the risk of ACT failure in sub-Saharan Africa. Additionally, *P. falciparum* parasites with deletions causing false negative rapid diagnostic test (RDT) results have been detected at high frequency in Eritrea[9,10] and Ethiopia[11,12]. The causal mutations underlying these phenotypes[12–16] and resistance to other common antimalarials are well characterised. By informing on the frequency and distribution of these mutations, genomic surveillance could play a crucial role crafting evidence-based policies to limit their spread and improve malaria control.

[1]Nuffield Department of Medicine, University of Oxford, Wellcome Centre for Human Genetics, Oxford, UK. [2]PATH, Lusaka, Zambia. [3]National Malaria Elimination Centre, Chainama, Lusaka, Zambia. [4]Max Planck Institute for Infection Biology, Berlin, Germany. [5]Imperial College London, London, UK. [6]Department of Pathology and Laboratory Medicine and Center for Computational Molecular Biology, Brown University, Providence, RI, USA. [7]These authors contributed equally: Mariateresa de Cesare, Mulenga Mwenda. ✉e-mail: hendry@mpiib-berlin.mpg.de

Despite its potential value, multiple challenges limit widespread genomic surveillance of *P. falciparum* malaria. First, the nuclear genome is 23 Mbp[17]—considerably larger than typical bacterial (-3–5 Mbp)[18,19] or viral genomes (-10–100 kbp)[20]. At present, this renders whole-genome sequencing strategies prohibitively costly to scale. Second, although targeted sequencing strategies—such as those employing multiplex polymerase chain reaction (PCR)[21–23], molecular-inversion probes[24,25] or hybrid capture — can be potentially more cost-effective, the genome of *P. falciparum* is extremely (A+T)-rich[17] and often there is little unique and biochemically-suitable sequence (e.g., for primer or probe design) within proximity of targets. This makes the development of these approaches particularly difficult for *P. falciparum*. Third, many regions with a high unmet need for *P. falciparum* genomic surveillance are in sub-Saharan Africa, yet most existing targeted sequencing approaches have been developed for Illumina platforms[21–23,25]. Due to their complexity, costs and maintenance requirements, these platforms are concentrated in centralised sequencing facilities—few of which are in sub-Saharan Africa. Although this situation is improving[26], deficits in local sequencing capacity still impel many small- and medium-sized labs to ship samples internationally for sequencing. This reduces country engagement, introduces ethical and logistical issues around sample export, and inevitably increases time to result, potentially delaying evidence-based policy decisions.

At the same time, there has been growing use of nanopore sequencing for pathogen genomic surveillance, facilitated by the small and portable MinION sequencing device (Oxford Nanopore Technologies). The MinION can be deployed in low resource settings, requires no maintenance, and permits real-time data analysis[27]. It has been successfully deployed during Ebola[28], Zika[29], and SARS-CoV-2 outbreaks[26]. A key advantage of nanopore-based sequencing is the generation of long reads (kbps to Mbps)[30] that can improve mapping and structural variant detection[31], while a disadvantage is a higher base-level error rate compared to instruments from Illumina or Pacific Biosciences (PacBio). Although important proof-of-principle studies have demonstrated the feasibility of nanopore-based sequencing of *P. falciparum*, and investigated the consequences of its higher error rate[32–34], comparatively little effort has been made to develop methods for routine nanopore-based genomic surveillance of malaria.

In this study, we developed a flexible and cost-effective approach to targeted *P. falciparum* sequencing using the MinION. Flexibility is created through the development of open-source software, called *multiply*, that enables multiplex PCR design for a user-defined set of target genes and/or regions across the *P. falciparum* genome. We use this software to create eight- and sixteen-target amplicon sequencing panels, which encompass genes associated with antimalarial drug resistance, RDT failure, complexity of infection (COI) inference and malaria vaccine target *csp*[35,36]. To sequence these panels we devised an optimised protocol that utilises dried blood spots (DBS) as input and costs approximately USD $25 per sample. We validate this approach on mock samples and Zambian field samples collected as DBS, and demonstrate adequate sequencing coverage of target genes, a high SNP calling accuracy within coding sequence (CDS), and how *P. falciparum* within-sample diversity is detectable in long-read data through analysis of the surface antigen gene *msp2*. Finally, we perform a proof-of-principle experiment demonstrating that our assay can identify *hrp2/3* deletions that cause false-negative RDT results, presenting a novel statistical model for deletion calling from amplicon sequencing data.

## Results

### Designing amplicon panels for *P. falciparum* with *multiply*

New amplicon sequencing panels require the development of a multiplex PCR which, even for a moderate number of targets, entails evaluating vast combinations of primers for off-target binding, primer dimers, or polymorphic sites in the study population. To facilitate this process for amplicon panels where the targets are distributed across larger genomes (i.e., in contrast to tiling PCR of smaller pathogen genomes[37]), we developed software called *multiply* (Fig. 1a). *multiply* provides a rapid and flexible approach to multiplex PCR design given a user-supplied list of target genes and/or regions. Briefly, *multiply* first generates a diverse set of candidate primers for each target using *primer3*[38]. It then searches for polymorphic sites within primer binding locations by intersecting them with user-supplied Variant Call Format (VCF) files; computes primer-dimer scores for all candidate primer pairs using an algorithm similar to that described by Johnston et al.[39]; and identifies potential off-target binding sites using *blastn* against the *P. falciparum* reference genome[40,41]. At present, *multiply* does not check for potential off-target binding sites in the human genome, or in the genomes of other blood-borne pathogens. Results from these three steps are combined into a cost-function that scores multiplex PCR primer combinations, with a lower score indicating a better predicted performance. Finally, the cost-function is minimised using a greedy search algorithm to identify optimal combinations of primers for the specified targets.

We used *multiply* to develop a multiplex PCR for *P. falciparum* malaria, selecting eight target genes that would maximise the public health utility of our data (Table 1). To leverage the long-read capability of nanopore sequencing, we restricted candidate amplicons to 3–4 kbp; aiming to produce CDS-spanning amplicons that would still being feasible for PCR. In the design process, *multiply* considered a total of 194 candidate primers across the eight targets. For these candidate primers, it identified 383 high scoring off-target complementary matches in the 3D7 reference genome (>12 bp aligned from the 3' end). Overall, 209 matches involved candidate forward primers for *dhps*; a candidate reverse primer for *plasmepsin I* (*pml*) had 35 matches; a candidate reverse primer for *kelch13* had 24 matches; and most other candidate primers had 5 or less matches. By comparing to the variant calls from 7113 *P. falciparum* whole genome sequences in the Pf6 data release[42], *multiply* identified 11 common SNPs (set to minor allele frequency >5% in any Pf6 population) within binding locations of candidate primers, which were excluded. Of the 18,915 unique pairwise alignments *multiply* computed between candidate primers, 585 had potentially problematic dimer scores (score < − 6). Using a greedy search algorithm, *multiply* heuristically minimised these factors to suggest a multiplex PCR primer combination from the over 370 million possibilities given the candidate primer set.

We call the amplicons produced from this multiplex PCR the NOMADS8 (**N**MEC-**O**xford **M**alaria **A**mplicon **D**rug-resistance **S**equencing) panel. In total, the amplicons cover 28.8 kbp with an (A+T)-composition of 79%. The full coding sequences for 7 of 8 gene targets are captured completely within their amplicons. *mdr1* has a coding sequence covering 4259 bp; our amplicon is only 3773 bp but includes important drug-resistance mutations (e.g., N86Y to D1246Y)[43]. Using PCR conditions with reduced annealing and extension temperatures[44], we were able to obtain robust amplification of all individual targets and produce bands consistent with expectation for the multiplex, as assessed by agarose gel electrophoresis (Supplementary Fig. 1).

We used *multiply* to expand the NOMADS8 panel to include an additional eight targets. These were *ama1*, a highly polymorphic gene used in COI estimation[45]; the RTS,S and R21 vaccine target *csp*[35,36]; and the RDT antigen genes *hrp2* and *hrp3*[16], as well as their flanking genes. To incorporate these eight targets, *multiply* considered an additional 214 candidate primers and, keeping the 16 primers of the NOMADS8 panel fixed, repeated the selection process described above. The resulting amplicon panel, called NOMADS16, covers a total of 54.7 kbp (Table 1).

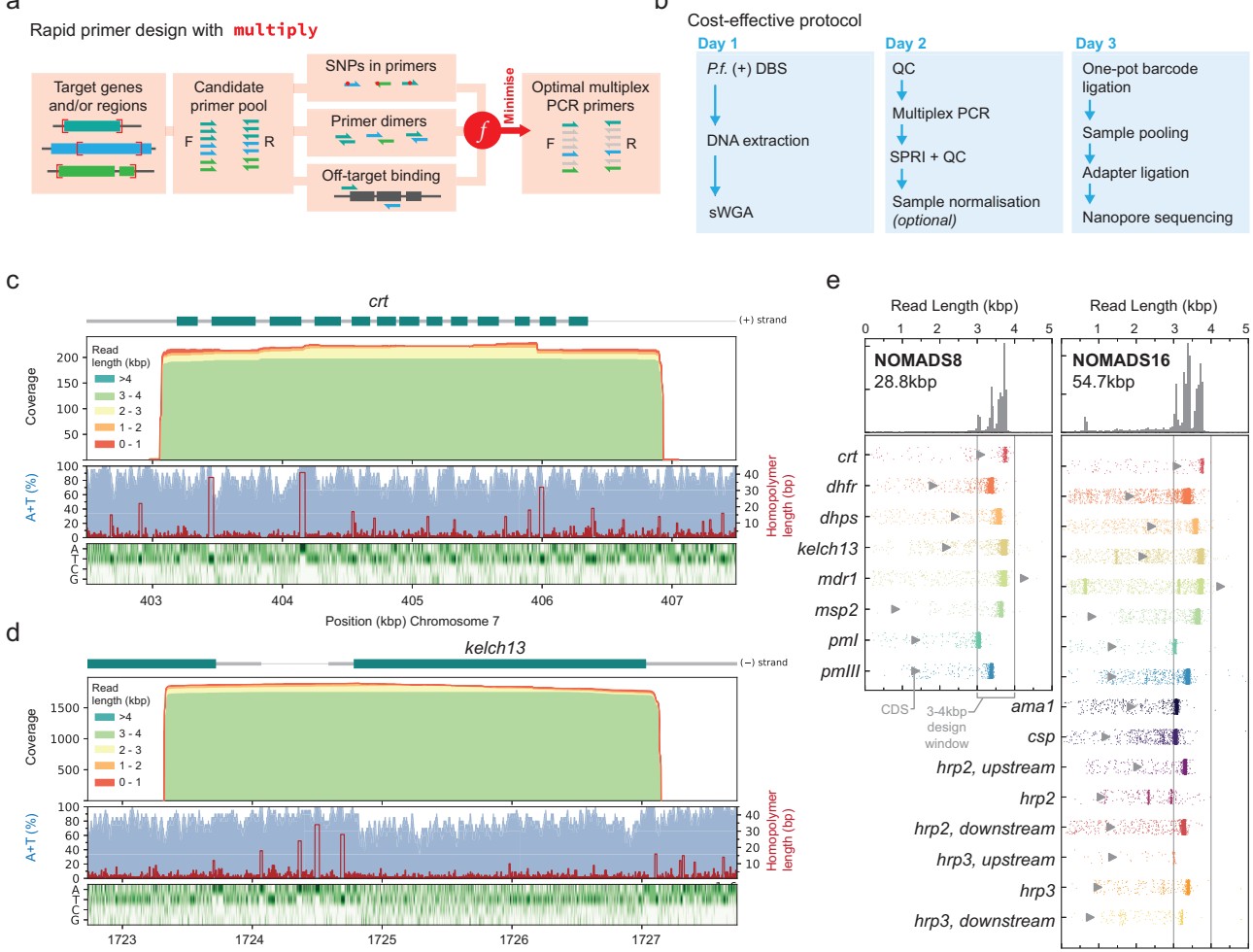

**Fig. 1 | Design of long-range multiplex PCRs for the low-complexity *P. falciparum* genome using *multiply*.** **a** Multiplex PCR primer design workflow by *multiply*. An optimal set of primers is selected from a large candidate pool; minimising SNPs in primer binding sites, primer dimers, and off-target primer binding with a cost function. **b** Schematic of a cost-effective protocol for targeted nanopore sequencing of *P. falciparum* malaria from dried blood spots (DBS) that takes three days and costs ~ USD $25 per sample. **c** Histograms of *crt* and (**d**) *kelch13* coverage stratified by read length. (A+T) percentage in 20bp sliding widows (blue) and homopolymer run length (red) are shown, as well as a heatmap of nucleotide composition. For both genes the entire coding sequence (CDS) is covered in the majority of reads. **e** Read length distributions for NOMADS8 (left, 28.8 kbp total) and NOMADS16 (right, 54.7 kbp total) amplicon panels. Grey triangle indicates coding sequence (CDS) length. Amplicons were designed to be 3–4 kbp. Marginal distribution for all amplicons displayed at top. Data for (**c**–**e**) are from a mock sample created from *P. falciparum* 3D7 and human DNA (Methods).

## Minimising *P. falciparum* amplicon sequencing costs on the MinION

We combined existing and novel optimisations to minimise the costs of *P. falciparum* target amplification and sequencing on the MinION (Fig. 1b). Briefly, our protocol starts with DBS as input for DNA extraction, which are relatively non-invasive and easy to collect. Bulk *P. falciparum* DNA is enriched with a reduced-volume selective-whole genome amplification (sWGA) step, saving approximately USD $4 per sample while still maintaining sufficient yield for subsequent multiplex PCR (Supplementary Fig. 2). Amplicons are barcoded and pooled using a modified version of a simple and cost-effective one-pot protocol[46]. Overall, the protocol from sample to sequence can be completed in 2–3 days at USD $25 per sample, assuming 96 samples are run on a R9.4.1 (FLO-MIN106D) or R10.4.1 (FLO-MIN114) MinION Flow Cell without washing (Supplementary Table 1). Smaller batches of 24 samples run on a Flongle Flow Cell (FLO-FLG001) add a negligible extra USD $1 per sample.

## Producing long-read data for policy-relevant *P. falciparum* genes

We explored the read lengths that are generated with our amplicon panels and protocol by sequencing a mock sample, created by combining *P. falciparum* 3D7 and human DNA in vitro, on a Flongle Flow Cell (Methods, Fig. 1c, d, e). For the NOMADS8 panel, the median length of reads that mapped to the *P. falciparum* reference genome and overlapped a target gene was 3.59 kbp. All eight target genes had a median read length greater than 3.04 kbp and, excluding *mdr1*, on average 91.7% of reads that overlapped a target gene spanned its entire CDS. This included reads spanning all 13 exons of *crt1* and the entire CDS of the artemisinin-resistance associated gene *kelch13* (Fig. 1c, d). In several cases, longer amplicons enabled *multiply* to select primers that bind to regions with more moderate (A+T) compositions in adjacent genes, and this was the case for the forward primer used to amplify *kelch13* (Fig. 1d). Similarly, the median length of target-overlapping reads for the NOMADS16 panel was 3.37 kbp, with an average of 88.2% of these completely spanning their target's CDS (excluding *mdr1*).

**Table 1 | Target genes for the NOMADS8 and NOMADS16 amplicon sequencing panels**

| No. | Reference ID | Gene Name | CDS | Amplicon | Relevance | Panel(s) |
|---|---|---|---|---|---|---|
| 1 | PF3D7_0709000 | *crt* | 3095 | 3874 | Chloroquine[77,78] | N8, N16 |
| 2 | PF3D7_0417200 | *dhfr* | 1826 | 3463 | Pyrimethamine[64,66,79] | N8, N16 |
| 3 | PF3D7_0810800 | *dhps* | 2416 | 3656 | Sulfadoxine[64,80,81] | N8, N16 |
| 4 | PF3D7_1343700 | *kelch13* | 2180 | 3826 | Artemisinin[13,14] | N8, N16 |
| 5 | PF3D7_0523000 | *mdr1* | 4259 | 3773 | Mefloquine[82] | N8, N16 |
| 6 | PF3D7_0206800 | *msp2* | 818 | 3720 | COI, recrudescence[50] | N8, N16 |
| 7 | PF3D7_1407900 | *pmI* | 1358 | 3101 | Piperaquine[83,84] | N8, N16 |
| 8 | PF3D7_1408100 | *pmIII* | 1355 | 3468 | Piperaquine[83,84] | N8, N16 |
| 9 | PF3D7_1133400 | *ama1* | 1868 | 3138 | COI[45] | N16 |
| 10 | PF3D7_0304600 | *csp* | 1193 | 3107 | Vaccine Target[35,36] | N16 |
| 11 | PF3D7_0831700 | *hsp70x* (*hrp2* up) | 2039 | 3366 | RDT[16] | N16 |
| 12 | PF3D7_0831800 | *hrp2* | 1063 | 3097 | RDT[16] | N16 |
| 13 | PF3D7_0832200 | None (*hrp2* down) | 1323 | 3333 | RDT[16] | N16 |
| 14 | PF3D7_1372000 | None (*hrp3* up) | 1370 | 3066 | RDT[16] | N16 |
| 15 | PF3D7_1372200 | *hrp3* | 976 | 3444 | RDT[16] | N16 |
| 16 | PF3D7_1372300 | None (*hrp3* down) | 771 | 3303 | RDT[16] | N16 |

For each target, columns give information about the coding-sequence (CDS) and amplicon lengths in base pairs (bp), the epidemiological relevance (COI, complexity of infection) and whether the target is in both the NOMADS8 (N8) and/or the NOMADS16 (N16) panel. The CDS length is measured as the distance between the start and stop codon, including intronic regions if present. '*hrp2* up' and '*hrp3* up' refer to targets upstream of *hrp2* and *hrp3*, respectively; wheres '*hrp2* down' and '*hrp3* down' refer to downstream targets.

## Characterising sequencing efficiency and coverage across mock and field samples

Sufficient coverage over target regions is a precondition for accurate variant calling and other downstream analyses. Whether this is achieved depends on the total sequencing throughput, the proportion of that throughput that is on-target (i.e., maps to the intended organism and regions), and how uniformly on-target throughput is distributed across the target regions and samples.

We characterised the coverage generated by our protocol by running experiments with both the NOMADS8 and NOMADS16 panels on two different sample sets. The first set included 24 mock samples, created in vitro from standard laboratory or cultured strains of *P. falciparum* malaria (Methods, Supplementary Table 2). The second was a set of 28 DBS assessed as *P. falciparum* positive by RDT, and collected from a clinical setting in Kaoma, Zambia (Methods).

We sequenced the mock samples with the NOMADS8 panel on a Flongle Flow Cell, generating 345 thousand reads or 1.08 Gbp (Fig. 2a). Of these reads, 80.0% passed the Guppy quality control filter and had identifiable sample barcodes on at least one end. We mapped these reads to the *P. falciparum* 3D7 reference genome and found that 76.2% (61.4% of the total reads) mapped successfully. To understand the causes of mapping failure, all unmapped reads were subsequently mapped to the human reference genome. Nearly all of the reads failing to map to the *P. falciparum* genome mapped successfully to the human reference (99.5%). These reads tended be shorter and of lower quality than those mapping to *P. falciparum*, and in optimisation experiments we were able to reduce them by using a higher stringency DNA size selection step after adapter ligation (Supplementary Fig. 3, Methods). The human-mapped reads remaining in this experiment (18.5% of total) were not removed by size selection, despite being shorter. Of the reads mapping to *P. falciparum*, 93.5% mapped to target regions, suggesting that *multiply* largely avoided the production of off-target amplicons. In the end, 57.4% of total sequencing reads were on-target for this experiment. A similar percentage was found to be on-target for the NOMADS8 panel when sequencing field samples using a standard MinION Flow Cell (62.1%).

Next, we interrogated how uniformly on-target sequencing coverage was distributed across the amplicons of the NOMADS8 panel by quantifying the number of reads that overlapped each of our targets. For the mock and field samples, the median fold-difference in coverage between the highest and lowest abundance amplicons were 9.3 and 16.2, respectively (Fig. 2b, c). With both the mock and field sample sequencing runs, the rank-order of amplicons by abundance was consistent (mock samples, Spearman's $\rho = 0.77$; field samples, Spearman's $\rho = 0.85$; Supplementary Fig. 4a, b). This indicates that coverage variation across amplicons is largely systematic, and likely a function of differences in PCR efficiency, rather than stochastic. However, comparing mock and field sample sets, the order of amplicon abundances differed slightly, indicating sample-set dependent effects. For example, *mdr1* was lower abundance and *dhfr* was higher abundance in field samples; but notably, *mdr1* is present in multiple copies in the laboratory strain Dd2, which is used in 8 of 24 of our mock samples (Supplementary Table 2). *crt1* was the lowest abundance for both mock and field samples; also being the longest amplicon in the NOMADS8 panel (3874 bp) with the second highest (A+T) composition (81.62%, behind *pmIII* with 81.66%) and most bases in long homopolymers (605 bp in homopolymers length 4 or greater). Despite this, *crt1* still had a median of 230-fold coverage in the mock sample experiment and 508-fold coverage with the field samples (Supplementary Fig. 4a, b).

The NOMADS16 panel had less uniform coverage across amplicons in comparison to the NOMADS8 panel (Supplementary Fig. 5). In particular, the fold-difference between the highest and lowest abundance amplicons was 141 for the mock samples and 324 for the field samples. This was driven in part by the *hrp3* upstream amplicon producing very low median coverage relative to other amplicons in both experiments (mock samples, median of 28-fold coverage; field samples, median of 30-fold coverage; Supplementary Fig. 4c, d); with the *hrp3* upstream amplicon excluded, the fold-differences between the highest and lowest abundance amplicons was substantially reduced, but still higher than with the NOMADS8 panel (mock samples, 43.8; field samples, 51.5). As with the NOMADS8 panel, amplicons in the NOMADS16 panel had consistent relative abundances (mock samples, Spearman's $\rho = 0.85$; field samples, Spearman's $\rho = 0.84$ and Supplementary Fig. 4c, d), but again the specific ordered varied somewhat between mock and field samples.

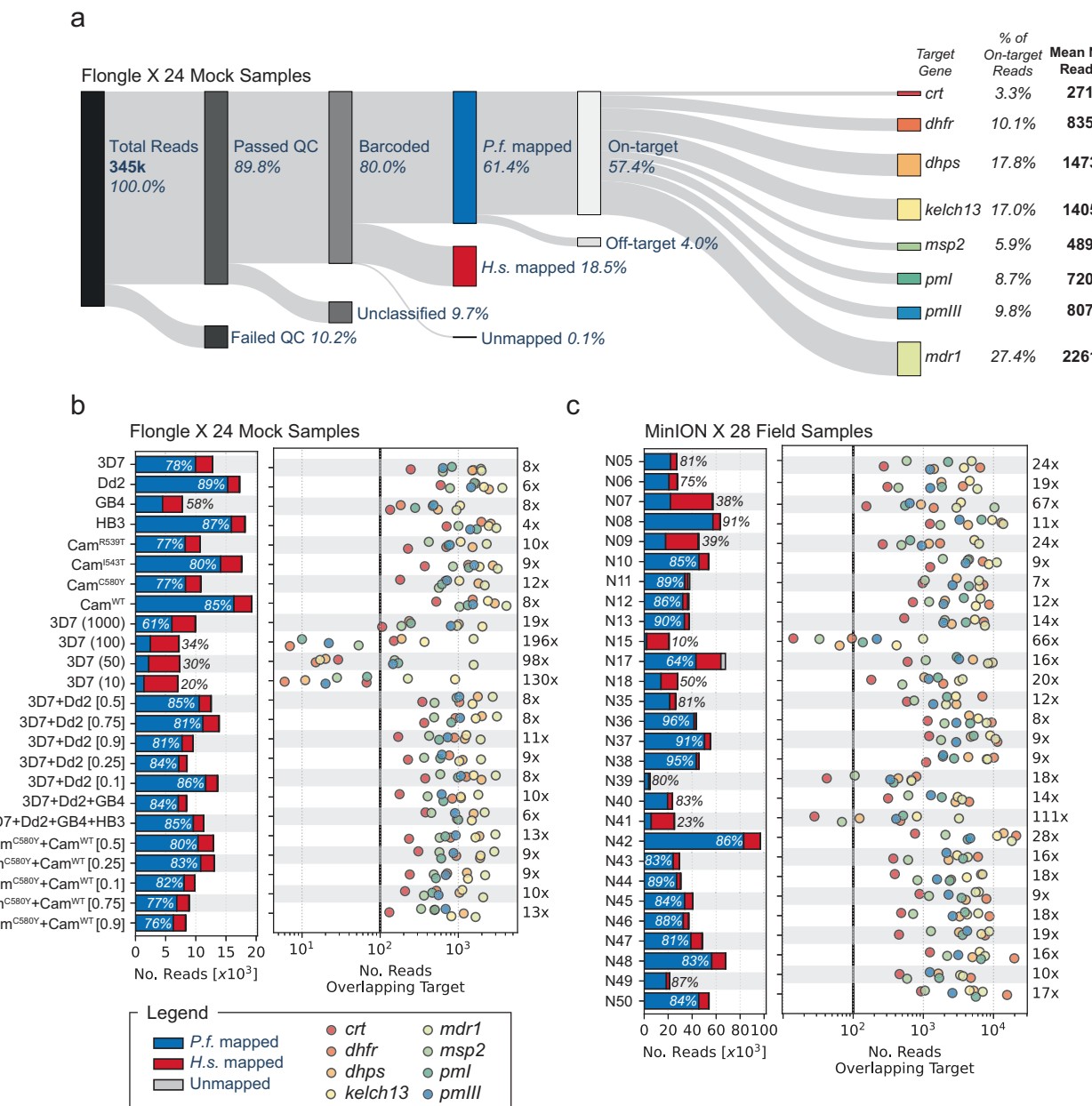

**Fig. 2 | Sequencing throughput and coverage across samples and target genes for the NOMADS8 panel. a** Diagram of reads produced on a Flongle Flow Cell (FLO-FLG001) sequencing 24 mock samples comprised of *P. falciparum* and human DNA. The leftmost bar represents all reads ($n = 345, 457$; 100%) generated during the sequencing run that are sequentially subdivided in the data analysis process to the reads of interest, i.e., those mapped to target genes ($n = 198, 030$; 57.4%). **b** Bar plot (left pane) displays the total number of reads generated for each sample stratified by mapping outcome: mapped to *P. falciparum* (*P.f.*) (blue), human (*H.s.*) (red), or failing to map (grey, too few to be visible). *P.f.* mapping percentages indicated with text. Scatter plot (right pane) displays the number of reads overlapping each target gene (labelled by colours) after mapping for each sample. Note number of reads (x-axis) is displayed in log-scale. For most samples, all target genes have >100x coverage. Number at right (e.g., 8x for 3D7) gives the fold-difference between the highest coverage and lowest coverage target. **c** Same as (**b**) but for 28 field samples collected as DBS from Kaoma, Zambia.

## Effect of parasitemia on sequencing performance

We examined the effect that sample parasitemia had on three measures of sequencing performance: the number of reads generated per sample, normalised to the mean for the sequencing run; the percentage of those reads that mapped to *P. falciparum*; and the fold-difference in coverage between the highest and lowest abundance amplicons for the sample (Fig. 3). We jointly analysed data from across six different sequencing experiments to take into account batch effects caused by technical factors or variation in sample quality. These experiments used both NOMADS8 and NOMADS16 and included three different types of sample sets: 120 mock samples created by combining *P. falciparum* and human genomic DNA at different ratios to replicate varying parasitemia; 28 field samples sequenced in Oxford, UK as part of a training; and 41 field samples sequenced from a governmental container lab in Lusaka, Zambia (Methods). Both sets of field samples were collected as DBS. Jointly these sample sets had parasitemia values ranging from 10 parasites per microlitre (p/µL) to over 100,000 p/µL. Unfortunately, we note that parasitemia data was missing for 12/28 field samples sequenced in Oxford.

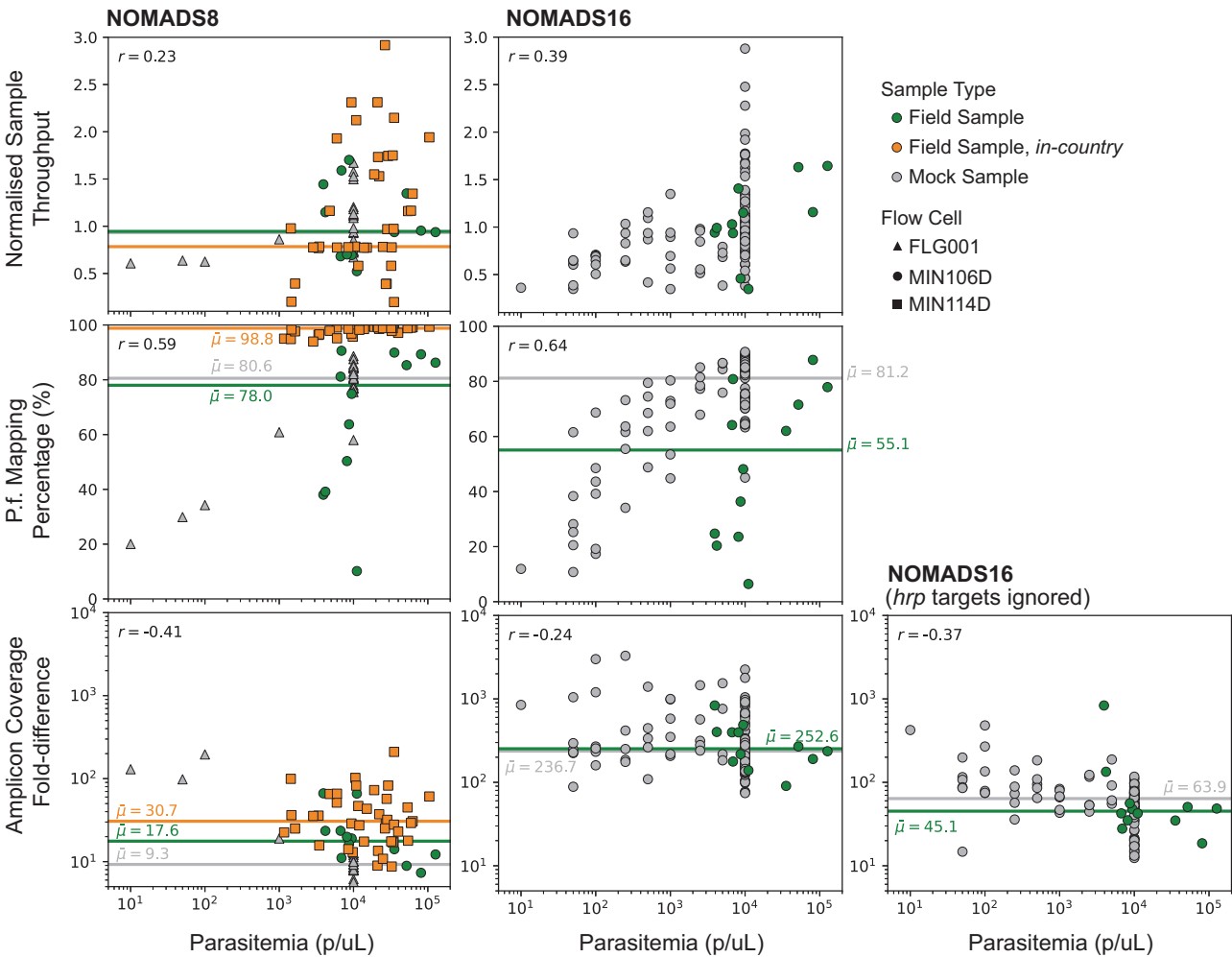

**Fig. 3 | Effect of parasitemia on sequencing performance measures.** Scatter plots display the effect that parasitemia (x-axis) has on the NOMADS8 (left), NOMADS16 (middle) and NOMADS16 amplicon panel with *hrp* genes ignored (right). Three measures of sequencing performance are shown (y-axis): "Normalised Sample Throughput", which is the number of reads generated for a sample, divided by the mean number of reads per sample for the sequencing run (top row); "*P.f.* Mapping Percentage", which is the percentage of all reads from a sample that mapped to *P. falciparum* (middle row); and the "Amplicon Coverage Fold-difference" which, for a given sample, is the ratio of the number of reads overlapping the highest abundance amplicon, divided by the number of reads overlapping the lowest abundance amplicon (bottom row). Each point is either an mock sample (grey), or a field sample sequenced in Oxford (green) or Zambia (orange). Samples sequenced on a R9.4.1 Flongle Flow Cell (FLG001) are indicated with triangles; R9.4.1 MinION Flow Cell (MIN106D), with circles; R10.4.1 MinION Flow Cell (MIN114D), with squares. Median values are shown as horizontal lines and Pearson correlation coefficient is given in top left. Note that parasitemia data is missing for 16 field samples.

We did not perform any sample input normalisation and found that the number of reads per sample had only a weak positive correlation with parasitemia for both the NOMADS8 (Pearson's $r = 0.23$) and NOMADS16 panels (Pearson's $r = 0.39$). The *P. falciparum* mapping percentages had a stronger positive correlation with parasitemia (NOMADS8, Pearson's $r = 0.59$; NOMADS16 Pearson's $r = 0.64$); values were markedly lower below approximately 1000 p/μL.

The coverage fold-difference across amplicons was higher at lower parasitemia values, producing a negative correlation that was more pronounced for the NOMADS8 (Pearson's $r = -0.41$) than the NOMADS16 panel (Pearson's $r = -0.24$). With the *hrp2/3* targets and their flanking genes removed, the fold-difference in coverage across the NOMADS16 panel was 5-fold less and the negative trend with parasitemia stronger (Pearson's $r = -0.37$). In addition to the *hrp3* upstream target being low abundance, several of the titrated mock samples contained *P. falciparum* laboratory strains Dd2 and HB3, which have *hrp2* and *hrp3* deletions, respectively. This partially masked the effect of parasitemia and increased variation in coverage. For both NOMADS8 and NOMADS16 panels, roughly 1000 p/μL was the threshold below which coverage variation across amplicons increased.

## SNPs are called accurately within coding sequences for clonal infections

We sought to assess how accurately molecular markers of antimalarial drug resistance could be detected with our method. Using *Clair3* to call variants[47], we examined SNP calls for set of substitution mutations associated with drug resistance (documented by the World Health Organisation[43]) across seven of our clonal mock samples that were sequenced on an R10.4.1 Flow Cell with a MinION Mk1b device (Fig. 4a). For the three mock samples containing *P. falciparum* laboratory strains, we identified all expected mutations and no false positives. Similarly, for the four mock samples created from cultured *P. falciparum* strains from Cambodia with documented artemisinin resistance, we identified the expected *kelch13* mutations and no false positives.

We expanded our analysis to examine SNP calling performance beyond known drug-resistance associated mutations and also characterised the effect read depth had on accuracy. We focused on the laboratory strains Dd2 and HB3, for which high-quality whole-genome assemblies exist[48]. For these two mock samples we randomly down-sampled the reads mapping to each target to sets ranging from 100 to

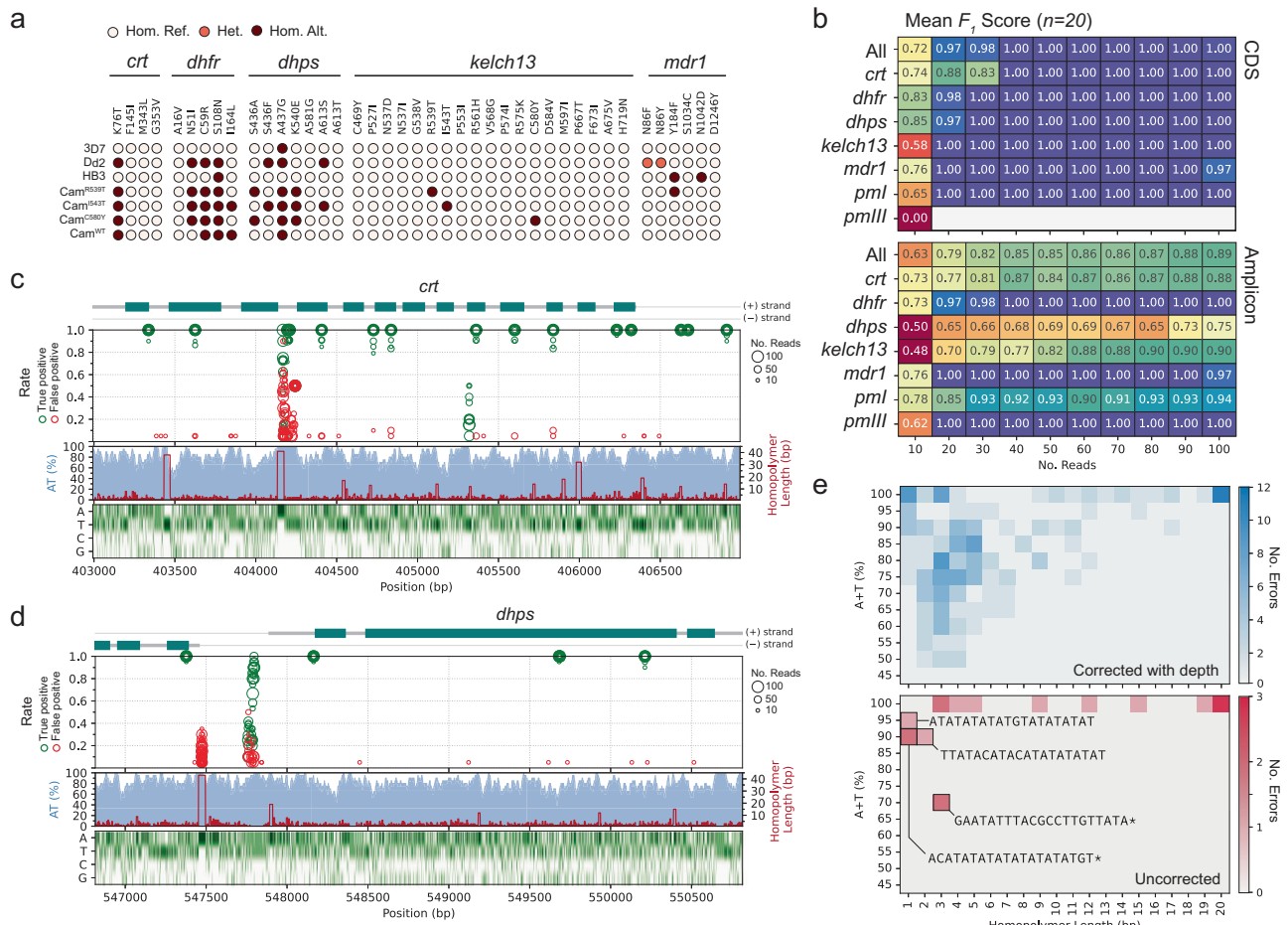

**Fig. 4 | SNP calling accuracy for a set of clonal mock samples. a** Genotyping results from *Clair3* for seven clonal mock samples and across 41 antimalarial resistance-associated mutations. Samples were sequenced with a R10.4.1 Flow Cell on a MinION Mk1b device. **b** Mean $F_1$-Score (harmonic mean of precision and recall) of SNP calling compared to PacBio data from for Dd2 and HB3 mock samples randomly downsampled to different read depths. Each square gives the mean $F_1$-Score across twenty in silico replicates (ten replicates for each of Dd2 and HB3) at the indicated read depth (columns) and across the indicated region (rows). In total, there are 200 in silico replicates across all depths. Top panel is limited to coding sequence ("CDS") and bottom panel the entire span of the amplicons ("Amplicon"). **c** Visualisation of true positive and false positive rate of sites spanning the *crt* amplicon in chromosome 7. From top to bottom, panels show an exon diagram of

*crt*; the true positive rate (green) and false positive rate (red) of each site across twenty replicates at a given read depth (indicated by circle size); A+T% in 20bp sliding windows (blue shade) and homopolymer length (red line); and heatmap of nucleotide composition. **d** Same as (**c**) but for *dhps* amplicon. **e** Heatmaps showing measures of sequence complexity in 20 bp windows surrounding sites where errors were observed. Rows indicate A+T% of the 20 bp window, columns indicate length of the longest homopolymer within the 20 bp window and colour gives number of errors. Top panel shows errors which were corrected with additional read depth (i.e., exist at depth < 100); bottom panel shows errors that persist at a depth of 100 reads. Selected sequences are shown; asterisk (*) marks sequences that are an example from a bin with greater than one sequence.

10 reads. We created replicates by repeating this procedure 10 times, thereby producing a total of 100 mock samples in silico with varying read depths for both Dd2 and HB3 (Methods). For these replicates we called variants using *Clair3* and treating the whole-genome assemblies as truth, evaluated accuracy using the haplotype comparison tool *hap.py*[49] (Methods). In Fig. 4b we show the mean $F_1$-scores (the harmonic mean of the precision and recall) for each target at a given read depth. The target *msp2* has been excluded as its very high sequence divergence from the reference genome makes it a case that should be handled separately, with a reference-free approach.

First we examined the coding sequences (CDS) of our targets (totalling 14.4 kbp, excluding *msp2*), as these are higher complexity and are also expected to capture the overwhelming majority of possible functional mutations. Overall, increasing from 10 to 30 reads resulted in a substantial improvement in the mean $F_1$ score ($F_1 = 0.72$ at 10 reads, to $F_1 = 0.98$ at 30 reads). With 40 reads or greater, SNPs within the CDS were called perfectly for all targets and replicates ($F_1 = 1.0$), aside from in a single replicate of *mdr1*. This error was in a Dd2 in silico

replicate at the N86 codon position. Clones of Dd2 have been observed to carry multiple copies of *mdr1*, inducing heterozygosity at codon N86 (AAT), as different copies carry N86Y (TAT) or N86F (TTT). In the complete set of reads overlapping *mdr1* in our Dd2 mock sample, the N86F mutation had a within-sample allele frequency of 67.4% (5451/8078 reads), most consistent with the mutation being carried by 2 of 3 *mdr1* copies. *Clair3* assumes a diploid genome, and this deviation from a 50% within-sample allele frequency likely led to the error. We observed similar errors, but at a higher frequency, in a previous analysis using an R9.4.1. Flow Cell (Supplementary Fig. 6).

We next expanded the analysis to the entire region spanned by our amplicons (totalling 25.2 kbp, excluding *msp2*), which includes 10.8 kbp of very low complexity (86% A+T) intergenic sequence. Here, SNP calling accuracy across our targets improved considerably with increasing numbers of reads, from an $F_1$-score of 0.63 overall with 10 reads, to a final $F_1$-score of 0.89 with 100 reads. We observed considerable variation in $F_1$-score across targets, which we hypothesised was due to differing amounts of low-complexity intergenic sequence.

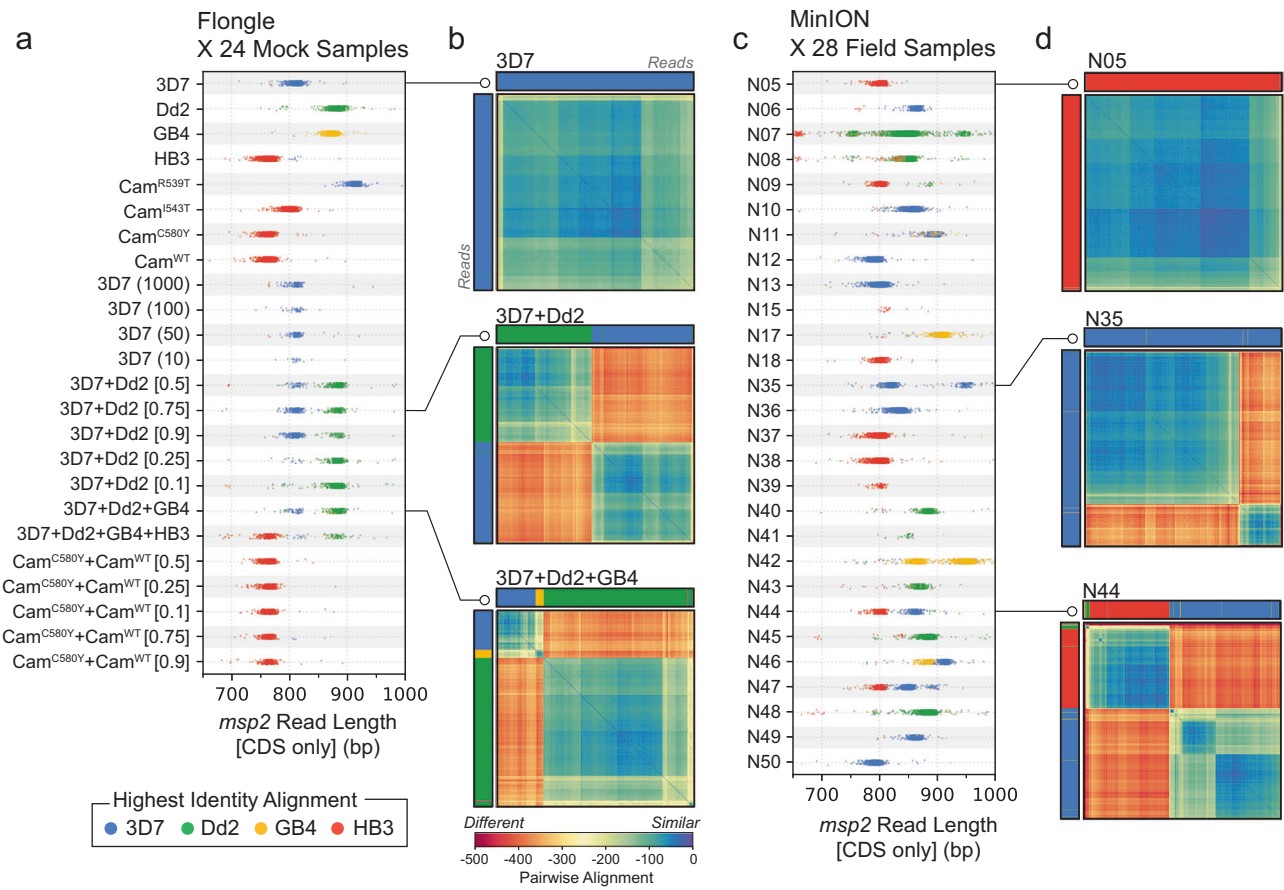

**Fig. 5 | Analysis of length polymorphism and nucleotide identity of *msp2*-derived reads. a** Read length distributions of *msp2* alleles across 24 mock samples. Each dot represents the length of a single read that was trimmed to the extent of *msp2* coding-sequence (CDS) after mapping. Individual reads are coloured by the laboratory strain to which they have the highest identity alignment. Multi-modal distributions are indicative of mixed infections. **b** Hierarchically clustered heatmaps of *msp2*-derived reads showing pairwise alignment scores. Each cell is coloured by the global pairwise alignment score between two *msp2*-derived reads, which have been hierarchically clustered along both rows and columns. Colours of

rows and columns indicate the laboratory strain to which each read has the highest identity alignment, as in (**a**). Heatmaps are shown for two different mock samples: clonal 3D7 (top); mixture of 3D7 and Dd2 (middle); and mixture of 3D7, Dd2 and GB4 (bottom). Note how reads cluster based on allele type. GB4 reads are under-represented in the bottom heatmap, likely due to lower DNA quality. **c**, **d** are the same as (**a**, **b**), but for 28 field samples from Zambia. In (**c**), read lengths distributions suggest the presence of both clonal and mixed infections. In (**d**), examples of likely clonal infection (top); two-strain infection (middle); and three-strain infection (bottom).

To evaluate this further, we visualised the genomic positions of erroneous SNP calls at different read depths across our target panel (Fig. 4c, d). Consistent with our hypothesis, we observed that areas with a high false positive rate, or diminished true positive rate, tended to be in very low complexity intergenic sequence. For example, between exons 3 and 4 of *crt* there is a homopolymer of 41 A nucleotides, around which SNP errors clustered (Fig. 4c). Similarly, upstream of *dhps* there is a 50 bp AT dinucleotide repeat region in which SNP errors were concentrated (Fig. 4d).

Finally, we systematically evaluated the sequence context of all unique sites where a SNP calling error was observed in any of the 200 in silico replicates in our analysis. These error-producing sites were divided into two groups: those in which the error was corrected with additional reads ($n = 198$), and those in which the error remained even in replicates with 100 reads ($n = 17$). We then computed the (A+T)-content and maximum homopolymer length in a 21 bp window centred on each of the sites (+/-10 bp). We found that sites where the SNP calling error could be corrected with additional read depth had lower (A+T)-content (mean 82.5% vs 94.4%) and shorter homopolymers in their flanking bases (mean 5.7 bp vs 8.3 bp) than the uncorrected SNP calling errors (Fig. 4e). Of the uncorrected SNP calling errors, 11/17 (65%) were situated in 21 bp windows consisting of

only A or T nucleotides and 6 of these contained homopolymers of length 10 or greater.

## Long-read sequencing of the surface antigen gene *msp2* provides insights into within-sample diversity

Long reads can facilitate interrogation of more complex regions of the genome. Both the NOMADS8 and NOMADS16 panel include the highly diverse surface antigen gene *msp2*, canonically used both for COI estimation and for distinguishing recrudescence from reinfection[50]. Critically, *msp2* genetic variation induces length polymorphism across a set of known repeat-containing alleles, enabling allele typing via capillary or gel electrophoresis.

We analysed reads deriving from *msp2* across our mock samples and observed length polymorphism analogous to that detected with electrophoresis approaches (Fig. 5a). To further characterise *msp2*-derived reads, we mapped them to each of the four *P. falciparum* laboratory strains used in our mock samples and labelled them by the strain to which they had the highest identity alignment (Methods). With this basic approach to allele classification, we could both confirm that the observed length polymorphism was driven by different underlying alleles of the expected types, and identify mock samples carrying multiple alleles. Next, we sought to explore an approach to

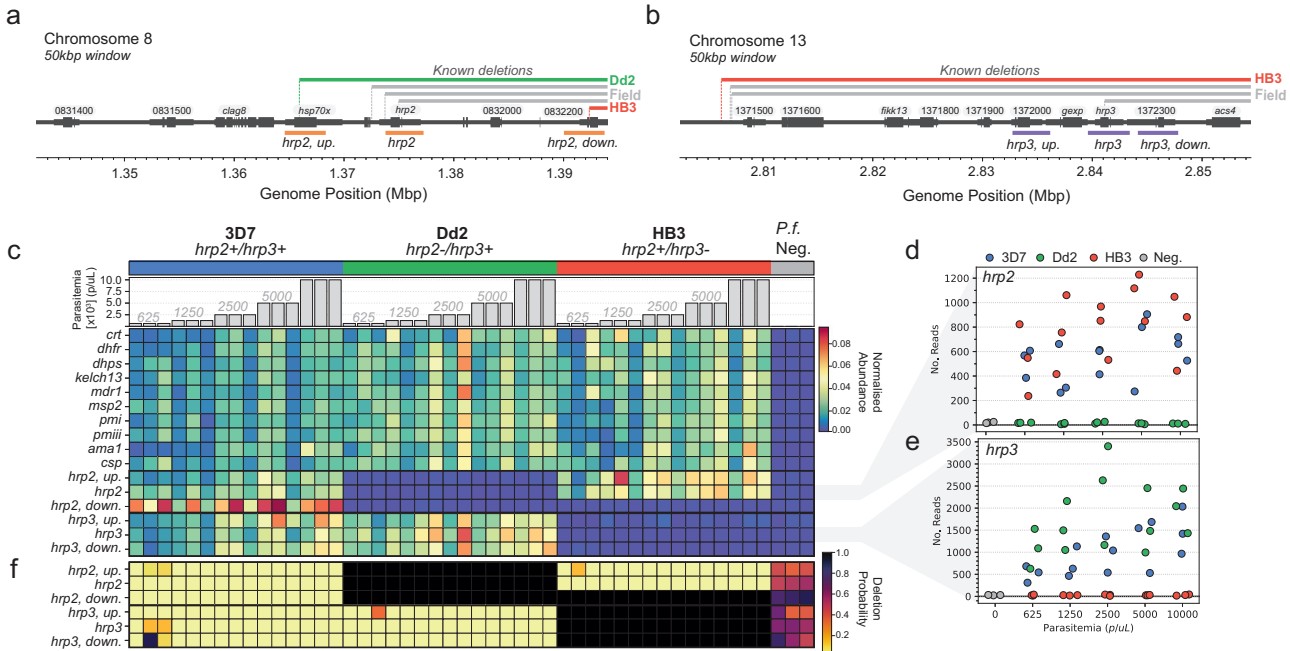

**Fig. 6 | Validation of *hrp2/3* deletion detection using the NOMADS16 panel.**
**a** Diagram showing the location of the *hrp2*, *hrp2* upstream and *hrp2* downstream amplicons in the NOMADS16 panel, within a 50 kbp window of chromosome 8. The chromosome is represented by a dark grey horizontal line, on which thicker segments demarcate genes (labelled above) and their exons. The genomic extent of documented *hrp2* deletions is displayed above the chromosome for lab strains Dd2 and HB3, and for a selection of three field strains[12,67]. Amplicon positions are shown below in orange. **b** Same as in (**a**) but for *hrp3* upstream, *hrp3* and *hrp3* downstream amplicons, shown in purple. Note Dd2 does not have a deletion within this window. **c** Heatmap displaying the normalised abundance of NOMADS16 panel amplicons (rows) across 48 mock samples (columns). The *P.f.* strain used in the mock sample (3D7, blue; Dd2, green; HB3, red; *P.f.*-negative, grey) and its parasitemia is indicated above the heatmap. The bottom six rows of the heatmap show

amplicons designed for detection of *hrp2* and *hrp3* deletions. **d** Scatterplot showing the relationship between amplicon abundance (y-axis, in number of reads) and parasitemia (x-axis) for the *hrp2* amplicon across all 48 mock samples. As in (**c**) the colour of points indicates the *P.f.* strain used in the mock sample. **e** Same as (**d**) but for the *hrp3* amplicon. **f** Heatmap displaying the probability of deletion for each of the six amplicons designed to support *hrp2/3* deletion detection (rows) across the 48 mock samples (columns) indicated in (**c**). Probabilities were calculated using a statistical model (Methods) that leverages all sixteen amplicons in NOMADS16 and estimates barcode misclassification/contamination rates from *P.f.*-negative samples. The expected deletions are detected with a very high degree of certainty (black squares). Uncertainty about *P.f.*-negative samples (deletion probability between 0.2 and 0.8) is expected as they have very few reads.

read classification that avoided using a priori information about allele types. To this end, we implemented a global alignment algorithm for pairs of reads that used base-level quality scores to assess the likelihood that both reads derived from the same underlying haplotype sequence (Methods). Using this algorithm, we performed pairwise global alignment of all *msp2*-derived reads for each sample and hierarchically clustered the resulting pairwise alignment score matrices (Fig. 5b). In cases where a single *P. falciparum* strain was used to produce a mock sample, the pairwise alignment score matrices had little structure, consistent with a single *msp2* allele being present. In cases where multiple *P. falciparum* strains were combined to produce a mock sample, structure within the pairwise alignment matrices was consistent with multiple *msp2* alleles being present.

The analysis using mock samples highlighted two limitations of these approaches. First, a general limitation of using only a single locus to learn about COI is that strains within a mixed/polyclonal infection may share the same allele at that locus, leading to underestimation of COI. We observed this with mock samples of COI = 2 created by combining Cam^WT and Cam^C580Y cultured *P. falciparum* strains. Second, reads identified as deriving from GB4 were underrepresented in higher COI mock samples (Fig. 5b). This may be due to the GB4 genomic DNA we obtained being lower quality. Consistent with this, the clonal mock sample created from GB4 genomic DNA produced less reads compared with the other laboratory strains (Fig. 2b).

We applied these approaches to characterise the *msp2*-derived reads in our field sample set and observed a variety of patterns reflecting clonal and mixed infections (Fig. 5c, d).

## Detecting *hrp2/3* deletions with the NOMADS16 amplicon panel

To characterise the ability of the NOMADS16 panel to detect *hrp2* and *hrp3* deletions that cause false-negative RDT results, we created a set of 45 clonal mock samples with a range of parasitemia levels (625–10,000 parasites per µL) from the lab strains 3D7 (*hrp2+/hrp3+*), Dd2 (*hrp2−/hrp3+*) and HB3 (*hrp2+/hrp3−*). We included three mock *P.f.*-negative samples as negative controls, yielding 48 mock samples total. We sequenced all 48 samples on a single R10.4.1. Flow Cell using a MinION Mk1b device, generating 4.85 million reads or 11.52 Gbp of sequencing data, and resulting in a mean of 39,602 reads mapping to *P.f.* per sample after quality filtering and demultiplexing (range 15,352–96,892; excluding negative controls).

As with previous experiments, we observed considerable variation in the mean abundance of the different amplicons in the NOMADS16 panel. We standardised this variation by converting the amplicon abundance for each sample to a proportion with respect to the total abundance of that amplicon across all samples. In a heatmap of these proportions, the expected *hrp2* and *hrp3* deletions were clearly visible in Dd2 and HB3 mock samples, respectively (Fig. 6c). More specifically, the Dd2 mock samples displayed a reduced abundance of the *hrp2* upstream and *hrp2* amplicons relative to 3D7 and HB3. For the *hrp2* downstream amplicon, both Dd2 and HB3 had reduced coverage relative to 3D7; which is consistent with other studies that have observed a deletion in HB3 near the end of chromosome 8, but not affecting the *hrp2* gene[12]. In the HB3 mock samples, we observed a reduced relative abundance of the *hrp3* upstream, *hrp3*, and *hrp3* downstream amplicons (Fig. 6c). Interestingly, despite the

mean abundance of the *hrp3* upstream amplicon (20 reads, *n* = 45) and *hrp3* downstream amplicon (51 reads, *n* = 45) being much lower than the average across all amplicons (2156 reads, *n* = 720), these experimental results suggest they are still informative for deletion detection.

Next, we examined in more detailed the abundance of the two amplicons that span the full-length *hrp2* and *hrp3* genes (Fig. 6d, e). The mean abundance of the *hrp2* amplicon in *hrp2* + mock samples was 669 reads (range 237–1229, *n* = 30), compared to a mean abundance of 14 reads (range 6–26, *n* = 15) in *hrp2* − mock samples, and 20 reads in the *P.f.*-negative controls (range 15–25, *n* = 3). For the *hrp3* amplicon, the mean abundance was 1362 reads in *hrp3* + mock samples (range 310-3401, *n* = 30), 28 reads in *hrp3* − mock samples(range 12–43, *n* = 15), and 28 reads in *P.f.* - negative controls (range 23–32, *n* = 3). The mean abundance for both *hrp2* and *hrp3* declined with parasitemia, but we still observed order of magnitude differences in abundance between deleted and wild-type parasite strains at 625 p/μL. We note that reads observed in the *P.f.*-negative samples and those expected to carry deletions are likely the result of barcode misclassification and/or contamination, and has been observed by others[34].

Finally, as proof-of-concept, we developed a statistical model for *hrp2/3* deletion detection from data generated by the NOMADS16 panel. Importantly, we devised a model that can be calibrated to an individual sequencing run, leverages information across all amplicons in the NOMADS16 panel, and takes into account barcode misclassification and/or contamination; ultimately providing a probability of *hrp2/3* deletion for each sample within a rigorous statistical framework (Methods). Applied to the data described above, our model detected all of the expected *hrp2* and *hrp3* deletions with complete certainty to a precision of four decimal places (*p* = 1.0000) (Fig. 6f).

## Discussion

Though widely deployed for genomic surveillance of viral and bacterial pathogens, nanopore sequencing of *P. falciparum* malaria is relatively rare[32–34]. Here, we have developed an approach to targeted nanopore sequencing of *P. falciparum* malaria that is flexible and cost-effective. Our approach begins with DBS as input and can produce genomic data of public health significance in 2 to 3 days at approximately USD $25 per sample. Importantly, DBS collection requires only a finger prick and is done routinely by malaria control programs. A major challenge with using DBS is that the modest amount of DNA extracted is primarily derived from the human host[51]. Here we use selective-whole genome amplification (sWGA) to enrich for bulk *P. falciparum* DNA prior to multiplex PCR, a strategy that has been adopted in several other *P.f.* amplicon sequencing workflows[21,23], to improve PCR performance and consistency from DBS. Simultaneously, we have demonstrated that the cost of sWGA can be substantially reduced when combined with targeted sequencing, allowing for a protocol that is both robust and affordable.

In developing *multiply*, we have provided a general and principled solution to the design of multiplex PCRs for targeted sequencing. This will enable rapid creation and updating of amplicon panels for *P. falciparum*, as well as accelerate the creation of panels for other organisms in the future. The software is open-source and freely available, allowing teams to design panels addressing their specific research or surveillance questions. Using *multiply*, we produced two amplicon sequencing panels containing eight- and sixteen-targets, reflecting the major public health uses of genomic data: tracking resistance to various antimalarial drugs, monitoring the sensitivity of RDTs, understanding the diversity of malaria vaccine targets and assessing within-sample diversity (which can help discriminate recrudescence from reinfection, and gives some indication of local transmission intensity). In contrast to all existing *P. falciparum* targeted sequencing approaches, our panels generate amplicons between 3 and 4 kbp, thereby producing individual reads that span the entire CDS of nearly all of our target genes.

A current priority of *P. falciparum* genomic surveillance is tracking the *hrp2* and *hrp3* deletions that can cause false-negative RDT results and are jeopardising the over 300 million RDTs distributed annually[52]. While several well-validated PCR-based methods exist to these deletions[16,53–55], there are only a few examples of detection by amplicon sequencing[34], or incorporation into amplicon sequencing panels. A set of best-practices for detecting these deletions by PCR recommended that an assay should: (i) target full-length *hrp2* exon 2 and the exon 1/2 boundary, to ensure both complete and partial deletions of *hrp2* are detected; (ii) target at least two single-copy genes, to ensure sufficient amplifiable DNA is present; and (iii) target one or both of the flanking genes, which are also lost in most deletions observed to date[16]. The NOMADS16 multiplex PCR was designed to meet all of these recommendations, and here we have shown it is able to accurately detect deletions across a set of mock samples of varying parasitemia. In addition, we have developed a novel statistical model for deletion detection that rigorously handles contamination and variation in sample quality, two issues that can complicate interpretation of amplicon sequencing data[34]. Although further validation on field samples is necessary, this work represents an important initial proof-of-principle for this approach.

A limitation of our current method is that it has weaker performance on low parasitemia samples in comparison with other *P. falciparum* amplicon-based methods designed for short-read sequencing[21–23,56]. Parasitemia and DBS sample quality (characterised by factors like age, storage conditions, number of blood spots and spot size) will influence the maximum amplicon length above which PCR performance will suffer due to an insufficient concentration of template DNA molecules of an adequate length. In addition, PCRs with longer amplicons typically have reduced efficiency in comparison with shorter alternatives. The 3 to 4 kbp, CDS-spanning amplicons in our panel exhibited robust assay performance on mock and field samples with above ~1000 parasites per microlitre. Additional experiments, especially with field samples, are necessary to more confidently establish this threshold and determine the requirements our long-read amplicon panels put on DBS collection procedures and quality. It is likely the NOMADS panels are best suited to higher parasitemia, symptomatic or clinical cases, rather than lower parasitemia asymptomatic cases. An advantage of developing *multiply* is that, should it be necessary, we will be able to rapidly design new amplicon panels with shorter lengths (e.g., 1–2 kbp). Moreover, work on adaptive sampling of reads during nanopore sequencing has recently been applied to *P. falciparum* malaria[57], and variations of this approach may help recover better data from low parasitemia samples.

An important question not directly addressed by this study is how sensitively our assay can detect minor clones, and the mutations they carry, in mixed/polyclonal *P. falciparum* infections[58]. The extent to which a sequencing method can detect minor clones depends on two sequential processes. The first is the reliability with which the laboratory protocol recapitulates, in the sequencing reads, the number and proportions of *P. falciparum* strains that existed in the cognate sample. This is fundamentally a sampling process, with higher variation and lower sensitivity expected in low parasitemia and low coverage samples; but it is also influenced by the non-linear dynamics of any amplification procedures that may be employed. Our approach uses sWGA, which has been shown to weaken correlations between strain proportions in a mixed infection and within-sample allele frequencies[23]. Once reads from different clones have been generated, the second process is to identify them by variant calling and/or haplotype inference, simultaneously distinguishing true variation from error or contamination. In clonal samples we determined that 30- to 50-fold coverage is sufficient for high accuracy SNP calling across all but the most extreme low complexity stretches of the genome (i.e., 100% (A+T)-content and/or hompolymers > 10bp). The fact that coverage levels tens- to hundreds- of times higher than this can be readily

attained gives some indication that detection of minor clones comprising ~ 5 to 10% of the sample may be feasible. Critically, we used *Clair3* to perform variant calling, which was designed for diploid human or haploid genomes[47], but not for samples with varying and unknown ploidy as is the case with *P. falciparum*. In order to properly investigate the limits of minor clone detection, haplotype inference tools that can handle complex *P. falciparum* infections in conjunction with the greater length and higher error rate of nanopore reads must be developed. Going forward, these may be built by adapting existing short-read haplotype inference tools, such as *DADA2*[59], for nanopore; by adapting nanopore-based tools such as *Clair3*[47] or *WhatsHap*[60], for malaria; or be developed as new fit-to-purpose tools. *SeekDeep*[61], an algorithm that has been successfully optimized for PacBio reads[62], may be more readily adaptable to nanopore.

Long-read amplicon sequencing of *P. falciparum* malaria brings benefits for malaria genomic surveillance. There are three immediate examples. First, long reads, especially those spanning entire CDS, are better suited for the detection of rare and novel mutations. Approaches using smaller reads typically focus on ~ 250 bp regions around known, common mutations, and have primer binding locations within the target gene. Therefore, novel mutations can emerge undetected or disrupt primer annealing, ultimately requiring the redesign or expansion of an amplicon panel. For *P. falciparum* a critical surveillance region is the propeller domain of *kelch13*[63], which harbours an expanding list of mutations conferring artemisinin resistance[4,43], but at 855 bp is too long to capture with a single amplicon in short-read sequencing. Second, once suitable computational tools are developed, long reads will enable epidemiologically relevant read-based phasing of variants within target genes[60]. For example, pyrimethamine treatment failure is predicted on the basis of a triple mutation within *dhfr* including N51I, C59R and S108N[64]; however, in some geographies single, double, and triple mutations exist, complicating this prediction for mixed infections[65–67]. Third, longer reads allow for better mapping in structurally complex or repetitive regions of the genome, and can assist with structural variant detection[31]. The investigation of several control-relevant regions of *P. falciparum*, including *msp2*, the histidine-rich proteins *hrp2* and *hrp3*, and the vaccine target *csp*; will all benefit from long-read sequencing.

Over 95% of all *P. falciparum* malaria cases occur in Africa[3], and yet the vast majority of *P. falciparum* genomic data is generated elsewhere. This discrepancy has resulted, in part, due to a preponderance of protocols making use of second-generation sequencing platforms with inaccessibly high capital and maintenance costs. A flexible and cost-effective protocol for nanopore sequencing of *P. falciparum* malaria that uses the MinION significantly expands the settings in which genomic data collection is possible. While on-site or clinical sequencing remains impractical, there exists a multitude of research and public health laboratories across Sub-Saharan Africa with interest in generating *P. falciparum* genomic data who can benefit from our approach. It is important to highlight that challenges still exist for widespread implementation, in particular establishing timely, reliable, and affordable procurement processes for scientific reagents and equipment in Sub-Saharan Africa. Notwithstanding, the over 100,000 SARS-CoV-2 genomes sequenced on the African continent during the pandemic demonstrate that these challenges can be overcome[26]. Given the rapid ongoing spread and, in some cases even, confluence[68], of *P. falciparum* drug and diagnostic resistance mutations in Africa, now is a critical time to expand *P. falciparum* genomic surveillance on the continent.

## Methods

### Development of multiplex PCR panels
The NOMADS8 and NOMADS16 panels were generated using a beta version of *multiply*, called *pf-multiply*, available at: https://github.com/JasonAHendry/pf-multiply(design file for NOMADS8 and NOMADS16).

Both use a BED (*.bed) file to delineate the mdr1 amplicon. NOMADS8 was generated first using the command:

```
python multiply.py -d designs/pf-nomads8-mdr1part.ini
```

NOMADS16 was created by using the `augment` command of *pf-multiply*. The NOMADS8 multiplex PCR primers were combined at equimolar amounts for total primer concentration of 0.6 μM; after an initial sequencing run with mock samples on a R9.4.1 Flongle Flow Cell, primer concentrations were crudely adjusted (doubled or halved) based on observed amplicon abundances, keeping the total primer concentration fixed. The same procedure was repeated with NOMADS16; i.e., one round of primer concentration adjustment was performed.

### Creating mock samples of *P. falciparum* and human DNA
We ordered *P. falciparum* genomic DNA for laboratory strains 3D7, Dd2, GB4 and HB3 and Cambodian field derived strains IPC 5202 (*kelch13* R539T); IPC 4912 (*kelch13* I543T), IPC 3445 (*kelch13* C580Y); and IPC 3663 (*kelch13* WT)[63] from BEI resources (www.beiresources.org). To create 10,000 p/μl in vitro DNA mixtures we diluted these stocks to 0.25ng/μl in 25ng/μL human genomic DNA from a pool of 36 HapMap cell lines[69]. DNA mixtures were then combined at different numbers and ratios to replicate mixed infections of different proportions or COI, and/or serial diluted in additional human genomic DNA to replicate lower parasitemia infections (Supplementary Table 2). For validation of *hrp2/3* deletions, parasite lines 3D7 (NF54), Dd2 and HB3 were cultured a 5% hematocrit in commerical red blood cells obtained from DRK Blutspendedienst Nord-Ost gemeinnützige Gmb, as previously described[70]. Genomic DNA from all lines was extracted using a Qiagen Blood and Tissue Kit on parasite pellets lysed with 0.15% saponin. Extracted DNA was combined with human genomic DNA (Roche, 11691112001) to produce a 10,000 p/μl stock, as described above. Lower parasitemia strains were produced by 2-fold serial dilution of the 10,000 p/μl stock into human genomic DNA.

### Collection of field samples
Samples from Zambia are from two studies. The first were collected under an ethical waiver granted by the National Health Research Authority, Zambia under the Laboratory Quality Improvement Research In Ministry of Health Laboratories (NHRA000004/16/11/2021). Symptomatic patients visiting a clinic in Kaoma, Western Province, Zambia were diagnosed with an RDT while a microscopy slide and DBS (on Whatmann No3 filter paper) were also collected. All samples were de-identified and no demographic or clinicial data was recorded. Bulk DNA was extracted from DBS using a Qiagen QIAamp Kit following manufacturers instructions. Parasitemia was quantified by light microscopy from thin film blood slides. The second set were collected during a Therapeutic Efficacy Study (TES) conducted by the Ministry of Health through the National Malaria Elimination Centre under ERES Converge IRB under Therapeutic Efficacy Testing for Artemether-Lumefantrine, Artesunate-Amodiaquine and Dihydroartemisinin-Piperaquine in Selected Sites in Zambia. The TES study is routinely conducted to assess the efficacy of three ACT antimalarial drugs used to treat uncomplicated malaria. Symptomatic patients visiting a clinic in Solwezi district, North-Western Zambia, were diagnosed for malaria using an RDT while a microscopy slide and DBS were collected. Parasitemia was quantified for every positive RDT, and any patient with parasitemia of 1000 parasites/μl or higher was given the option to enrol in the study through the written consent process. Additional clinical information such as fever status and demographic data (i.e., age, height, weight) were collected. The patient's home address was recorded for study follow-up purposes, but the DBS were de-identified prior to any analysis being conducted.

## Laboratory protocol and sequencing

The complete laboratory protocol, including materials and primer sequences, is available online at protocols.io (https://www.protocols.io) as "Cost-effective targeted nanopore sequencing of P. falciparum malaria". In brief, for each sample 10–40 ng of extracted genomic DNA was used as template in a 50 μl sWGA reaction[51], but with reduced phi29 DNA polymerase (NEB #M0269S) to minimise costs. Afterwards, 2 μl of sWGA product was used as template in a 25 μl multiplex PCR with KAPA HiFi Polymerase (Roche #KK2101) and either the NOMADS8 or NOMADS16 primer pools. Multiplex PCR products were cleaned using a 0.5X ratio of NEBNext Sample Purification Beads (NEB #E7103) and eluted in 15 μl nuclease-free water. DNA elute was quantified using the Qubit (ThermoFisher #Q32854) and between 100 and 600 ng of DNA was taken forward for barcoding and sequencing. We ligate unique barcodes from an Oxford Nanopore Technologies (ONT) Native Barcode Ligation Sequencing Kit (SQK-LSK109 with EXP-NBD104, EXPNBD114 for R9.4.1 Flow Cells; SQK-NBD114.96 for R10.4.1 Flow Cells) to each sample using a modified one-pot barcoding protocol[46]. Samples are then pooled before adapter ligation and sequencing, where we follow ONT protocol recommendations.

## Bioinformatics pipeline

For experiments using R9.4.1 Flow Cells, FAST5 files generated by the *MinKNOW* software were basecalled using *Guppy* (v5.0.11) with a minimum quality score threshold of 8. For the Flongle experiment, we used the super-accurate (SUP) basecalling model and for all other experiments we used the high accuracy (HAC) basecalling model. For experiments using R10.4.1 Flow Cells, POD5 files were basecalled using *dorado* (v0.34; https://github.com/nanoporetech/dorado) using the super-accurate (SUP) model. FASTQ files were then demultiplexed using *Guppy* (v5.0.11), without setting the `--require_both_ends` flag, i.e., with single-end demultiplexing. Demultiplexed FASTQ files were mapped to release 52 of the *P. falciparum* 3D7 reference genome downloaded from *PlasmoDB*[71] (https://plasmodb.org) using *minimap2*[72] (v2.24-r1122) and the `-ax-ont` parameter setting. In the resultant BAM (*.bam) file, reads failing to map to the 3D7 reference were identified using *samtools*[73] (v1.16), with the command `samtools view -f 0x904`. These unmapped reads were converted back to FASTQ files using `samtools fastq` before being remapped to the GRCh38 human reference genome downloaded from NCBI's Genome Database (https://www.ncbi.nlm.nih.gov/genome) and subsequently excluded from downstream analyses. Reads deriving from targets were defined as those that overlapped the coding-sequence defined in the Gene Feature Format (GFF) (*.gff) for release 52 of the *P. falciparum* 3D7 reference genome downloaded from *PlasmoDB*[71]. Variant calling of reads mapping to the 3D7 reference genome was performed using the using the singularity image of *Clair3*[47] (v1.0.4; https://github.com/HKU-BAL/Clair3) in diploid mode with the flags `--platform='ont' --include_all_ctgs --enable_phasing` set. For the SNP calling analysis in Fig. 4, we sent all variants to the alignment model by setting `--var_pct_full=1.0` and `-ref_pct_full=1.0`.

## SNP calling accuracy analysis

**Downsampling reads.** We partitioned reads mapped to the 3D7 reference into those overlapping each of our target genes using *samtools*[73] (v1.16), thereby producing BAM files for each of our targets. For each target BAM, we downsampled reads using the `samtools view` command and `-s/--subsample` flag to achieve the desired number of reads. This procedure was repeated for the Dd2 and HB3 samples; downsampling to 10, 20, 30, 40, 50, 60, 70, 80, 90 and 100 reads for each target, and ten times for each number of reads. As a result, for each given depth and target, we produced 10 randomly downsampled BAM files for Dd2 and Hb3; for 20 replicates total. In Fig. 4b the 'All' category was created be concatenating the BAM files generated in this way for all targets in the NOMADS8 panel, excluding *msp2*.

**Creating a set of true variants.** Dd2 and HB3 have been sequenced to high depth on the Pacific Bioscience Sequencing SMRT technology and assembled, with resulting FASTA (*.fasta) sequences available on *PlasmoDB*[71]. To identify variants in these assemblies with respect to the 3D7 reference genome, we simulated high-quality (Phred 60) error-free reads in silico from the FASTA files, mapped them to the 3D7 reference genome with *minimap2* (v2.24-r1122), and then identified variants using the *bcftools*[74] (v1.16) `mpileup` and `call` commands. In particular, we simulated 60 error-free reads, half forward and half reverse strand, for each target in our NOMADS8 panel by extracting the FASTA sequence spanning +/-4kbp of the target, based on GFF files for Dd2 and HB3.

*Stratified variant call comparisons.* We used the tool *hap.py*[49](https://github.com/Illumina/hap.py) from a *Docker* image (jmcdani20/hap.py:v0.3.12) to compute measures of variant calling accuracy across different target regions in comparison to the true variant set described above. To restrict accuracy measure analysis to coding sequence, we subset the 3D7 GFF to only CDS features, identified the rows pertaining to our targets, and output the chromosome, start, and stop positions as a BED (*.bed) file. We used then used the `--stratifications` flag of *hap.py* to compute measures over these intervals. We used the annotated VCF (*.vcf) files produced by *hap.py* to generate positional plots of false- and true-positive rate across targets in Python.

## Analysis of *msp2* reads

**Computing coding sequence lengths.** After mapping reads to the 3D7 reference genome with *minimap2*[72] (v2.24-r1122), we extracted reads that completely overlapped the *msp2* (PF3D7_0206800) coding sequence using *bedtools*[74] (v2.31.0) with the `intersect -F 1.0` command. From the resultant BAM file, we trimmed these reads to the extent of the *msp2* coding sequencing by keeping only the section of each read that aligned within the interval [273689, 274507] of chromosome 2 (Pf3d7_02_v3); indels were retained if the bases on either side of them aligned within the interval. Unusually short trimmed reads (< 400bp) were removed as likely artefacts. Trimmed reads were used to create length distribution plots. They were independently mapped, using *minimap2*[72] (v2.24-r1122), to release 52 of the reference genomes for 3D7, Dd2, GB4, and HB3 downloaded from *PlasmoDB*[71]. We let *minimap2* output a PAF (*.paf) file and computed the identity of the mapping alignment by dividing column 10 (number of matches in alignment) by column 11 (total alignment length).

**Global pairwise alignment.** We implemented a banded version of the Needleman-Wunsch algorithm to compute global alignment scores between pairs of trimmed *msp2* reads. We parameterised the scoring model such that scores reflect the log-probability that the two observed reads derived from the same underlying haplotype sequence; i.e., that all alignment differences were caused by sequencing error. Assuming an indel rate of 5%, which is broadly consistent with observed error rates, we used a linear gap score of $log_{10}(0.05)$. For substitution scores, we took into account the base quality scores generated by *Guppy* as follows. Defining $x$ and $y$ as the two observed bases in the match, the likelihood that they were generated from the same haplotype base $h$ is

$$P(x,y|h,p_x,p_y) = \begin{cases} (1-p_x)(1-p_y) + \frac{p_x p_y}{3} & \text{if } x=y \\ p_x(1-p_y) + p_y(1-p_x) + \frac{2p_x p_y}{3} & \text{if } x \neq y \end{cases} \quad (1)$$

where $p_x = 10^{\frac{Q_x}{-10}}$ and $p_y = 10^{\frac{Q_y}{-10}}$, with $Q_x$ and $Q_y$ being the Phred-scaled base quality scores for $x$ and $y$. The substitution score is then computed as $log_{10}(P(x,y|h,p_x,p_y))$. For all alignments in this study, a band width of 80 bp centred on the diagonal of the global alignment

**Table 2 | Notation for *hrp2/3* deletion detection model**

| Symbol | Description | |
|---|---|---|
| $i$ | Sample index | $i \in \{1, 2, \ldots, n\}$ |
| $j$ | Target gene index | $j \in \{1, 2, \ldots, m\}$ |
| $\mathbf{X}_{n \times m}$ | Read count matrix | $x_{ij} \in \mathbb{Z}^+$ |
| $\mathbf{C}_{n \times m}$ | Copy-number matrix | $c_{ij} \in \{0, 1\}$ |
| $\delta$ | Indices of negative control samples | $\delta \subset \{1, 2, \ldots, n\}$ |
| $\phi$ | Indices of target genes without known deletions (i.e., non-*hrp*) | $\phi \subset \{1, 2, \ldots, m\}$ |
| $\mathbf{a}_n$ | Sample abundances | $a_i \in [0, 1]$ |
| $\epsilon$ | Read misclassification / contamination rate | $\epsilon \in [0, 1]$ |
| $\nu$ | Read count overdispersion term | $\nu \in \mathbb{R}+$ |
| $p_{ij}$ | Unadjusted library proportion of sample $i$ and target $j$ | $p_{ij} \in [0, 1]$ |
| $\pi_{ij}$ | Adjusted library proportion of sample $i$ and target $j$ | $\pi_{ij} \in [0, 1]$ |

matrix was used. Hierarchical clustering of the resulting scores was performed using the `scipy.cluster.hierachy.linkage` function from the *SciPy*[75] (v1.4.1) Python package.

## Statistical model for *hrp2/3* deletion detection

**Data and notation**. We first describe the data and notation used by our model (Table 2). Amplicon sequencing data is represented as a two-dimensional matrix of positive integers, **X**, which holds read counts after quality filtering, demultiplexing and mapping. The matrix **X** has rows representing samples, which are indexed by $i \in \{1, 2, \ldots, n\}$, and columns representing target genes, which are indexed by $j \in \{1, 2, \ldots, m\}$; each element, $x_{ij}$, represents the number of reads from sample $i$ that mapped to the target gene $j$. We define a corresponding binary matrix, **C**, where each element $c_{ij}$ indicates that the given target gene $j$ is either present ($c_{ij} = 1$) or deleted ($c_{ij} = 0$) in sample $i$. We define a vector **a** of size $n$ such that $a_i \in [0, 1]$, which represents the relative abundance of each sample in the sequencing library. Finally, we define two scalar parameters: a read misclassification rate, $\epsilon \in [0, 1]$, which represents the rate at which reads derived from sample $k$ contribute to another sample's read counts (i.e., to $x_{i \neq k, j}$), whether by contamination or incorrect sample assignment during demultiplexing; and a read count dispersion term $\nu \in \mathbb{R}+$, which is given a precise mathematical definition below.

**Model**. Each column of the read count matrix, $\mathbf{x}_j = \{x_{1j}, x_{2j}, \ldots x_{nj}\}$, contains the read counts for a given target gene $j$ across all $n$ samples. We model $\mathbf{x}_j$ with a Dirichlet-multinomial distribution:

$$P(\mathbf{x}_j; N_j, \alpha_j) = \frac{\Gamma(\sum_{i=1}^n \alpha_{ij}) \Gamma(N_j + 1)}{\Gamma(N_j + \sum_{i=1}^n \alpha_{ij})} \prod_{i=1}^n \frac{\Gamma(x_{ij} + \alpha_{ij})}{\Gamma(\alpha_{ij}) \Gamma(x_{ij} + 1)}, \quad (2)$$

where $N_j = \sum_{i=1}^n x_{ij}$ is the total read counts for target $j$ across all samples; and $\alpha_j = \{\alpha_{1j}, \alpha_{2j}, \ldots, \alpha_{nj}\}$ is a vector of compound parameters, $\alpha_{ij}$, for the target gene $j$. These $\alpha_{ij}$ determine the expected number of reads for each sample for a given target gene and are computed in three steps. First, we use the relative abundance of a sample, $a_i$, and its deletion status for target gene $j$, $c_{ij}$, to compute the expected proportion of reads generated for target gene $j$ that should derive from sample $i$:

$$p_{ij} = \frac{c_{ij} a_i}{\sum_{k=1}^n c_{kj} a_k}. \quad (3)$$

Note that $p_{ij}$ either equals zero, if target gene $j$ is deleted in sample $i$; or the proportion of sample $i$ in the sequencing library, but relative to only the samples where the gene is present. In the process of generating these reads, these expected proportions are altered through read misclassification and sample contamination, such that a different set of expected proportions, $\pi_{ij}$, are reflected in the final data. Here, we make the assumption that read misclassification happens at a fixed rate, $\epsilon$, and uniformly across samples, resulting in the expression:

$$\pi_{ij} = p_{ij}(1 - \epsilon) + (1 - p_{ij})\epsilon. \quad (4)$$

Both $p_{ij}$ and $\pi_{ij}$ sum to one for a given $j$. Finally, we parameterise the Dirichlet-multinomial with $\alpha_{ij} = \nu \pi_{ij}$. The effect is that the expected read counts, $x_{ij}$, equals the product of the total number of reads for target gene $j$, $N_j$, multiplied by the deletion-status and error-adjusted sample proportion, $\pi_{ij}$:

$$E[x_{ij}] = N_j \frac{\alpha_{ij}}{\sum_{k=1}^n \alpha_{kj}} = N_j \frac{\nu \pi_{ij}}{\sum_{k=1}^n \nu \pi_{kj}} = N_j \pi_{ij}. \quad (5)$$

The variance of $x_{ij}$ equals:

$$V[x_{ij}] = N_j \pi_{ij}(1 - \pi_{ij})\left(\frac{N_j + \nu}{1 + \nu}\right), \quad (6)$$

which increases as $\nu$ shrinks towards zero, or asymptotically approaches the variance of a binomial distribution, as $\nu$ grows towards infinity; $\nu$ controls read count overdispersion relative to a binomial distribution. In summary, the $\alpha_{ij}$ incorporate information about the deletion status of the target gene, the relative abundance of each sample in the library, the rate of misclassification in the sequencing run, and the amount of overdispersion in read counts across samples.

**Inference**. Our aim is to infer whether a target gene of interest is present ($c_{ij} = 1$) or absent ($c_{ij} = 0$) in a given sample, using all of the salient information in the read count matrix **X**. We approach this with a Bayesian formulation: treating the Dirichlet-multinomial distribution, described above, as the likelihood and computing the posterior probability over $\mathbf{c}_j = (c_{1j}, c_{2j}, \ldots, c_{nj})$ as:

$$P(\mathbf{c}_j | \mathbf{x}_j; \epsilon, \nu, \mathbf{a}) \propto P(\mathbf{x}_j | \mathbf{c}_j; \epsilon, \nu, \mathbf{a}) P(\mathbf{c}_j). \quad (7)$$

A natural choice of prior for each $c_{ij}$ would be a Bernoulli distribution, $c_{ij} \sim Bern(\theta)$, with $1 - \theta$ giving the expected probability of deletion. Here, for simplicity, we have chosen a uniform prior equivalent to $\theta = 0.5$, although in principle this could be adjusted based on previous knowledge about deletion prevalence of target gene $j$ in the regions from which the samples were collected.

Also for simplicity, we have chosen to treat $\epsilon$, $\nu$, and **a** as fixed parameters and we fit them using point estimation. Let $\delta$ be a set containing the indices of all the negative control samples included in the sequencing run, such that $|\delta|$ represents number negative controls. We first make a simple point estimate of the misclassification rate by taking the empirical mean of the $x_{ij}$'s for all these negative controls:

$$\epsilon = \frac{1}{|\delta| m} \sum_{i \in \delta} \sum_{j=1}^m x_{ij}/N_j. \quad (8)$$

This uses the fact that $E[x_{ij}]/N_j = \epsilon$ for cases where $a_i = 0$, which is true by definition for negative controls. Next we compute point estimates of the **a** and $\nu$ parameters. To make these estimates, we define a subset $\phi \subset \{1, 2, \ldots, m\}$ representing the indices of the target genes with no known deletions. In the context of the NOMADS16 panel, this includes ten target genes excluding *hrp2*, *hrp2, up.*, *hrp2, down.*, *hrp3*, *hrp3, up.*, and *hrp3, down*. Closed form maximum-likelihood estimators of the parameters of a Dirichlet or Dirichlet-multinomial do not

exist[76], and so instead we estimate $a_i$ using the empirical mean of $x_{ij}/N_j$ for the target genes in set $\phi$:

$$a_i = \frac{1}{|\phi|} \sum_{j \in \phi} x_{ij}/N_j. \tag{9}$$

Then, following Minka (2012)[76], we estimate $\nu$ using:

$$\log(\nu) = \frac{1}{|\phi|-1} \sum_{j \in (|\phi|-1)} \log\left(\frac{a_i(1-a_i)}{var(x_{ij}/N_j)} - 1\right). \tag{10}$$

With point estimates of $\epsilon$, $\nu$ and $\mathbf{a}$, we compute the posterior distribution of $\mathbf{c}_j$ using Markov Chain Monte Carlo (MCMC). For each target gene $j$, we run an independent Metropolis-Hastings algorithm to compute the posterior $\mathbf{c}_j$. We initialise the MCMC with $c_{ij}=1$ for all $i \in \{1, 2, \ldots, n\}$. In each iteration, we propose a new $\mathbf{c}'_j$ by choosing uniformly from $i$, and then switching the deletion status of the corresponding $c_{ij}$ by computing $c'_{ij}=1-c_{ij}$. As this proposal is symmetrical, the Hastings Ratio is 1 and we accept the update with probability:

$$\min\left[1, \frac{P(\mathbf{x}_j; N_j, \mathbf{c}'_j, \epsilon, \nu, \mathbf{a})P(\mathbf{c}'_j)}{P(\mathbf{x}_j; N_j, \mathbf{c}_j, \epsilon, \nu, \mathbf{a})P(\mathbf{c}_j)}\right] \tag{11}$$

In total we conducted 10,000 iterations of the MCMC for each target gene, discarding the first 500 as burn-in. Finally, the posterior probability that a given sample $i$ is carrying a deletion of target gene $j$ is equal to the fraction of the iterations in which $c_{ij}=0$.

### Reporting summary
Further information on research design is available in the Nature Portfolio Reporting Summary linked to this article.

## Data availability
Reads were mapped to release 52 of the *P. falciparum* reference genome for strains 3D7 (PlasmoDB-52_Pfalciparum3D7), Dd2 (PlasmoDB-52_PfalciparumDd2), GB4 (PlasmoDB-52_PfalciparumGB4) and HB3 (PlasmoDB-52_PfalciparumHB3) downloaded from *PlasmoDB*[71]; and to the GRCh38 human reference genome (GRCh38) downloaded from NCBI. Sequence data is available for download from NCBI's Sequence Read Archive (https://www.ncbi.nlm.nih.gov/sra) under the accession PRJNA956048.

## Code availability
*multiply* is available at: https://github.com/JasonAHendry/multiply. Bioinformatics pipeline, SNP calling accuracy, and *msp2* analysis code is available at: https://github.com/JasonAHendry/nomadic2.

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

## Acknowledgements

The NOMADS project is funded by the Bill & Melinda Gates Foundation (INV-003660 to K.S., D.J.B., M.M. and J.A.H., INV-048316 to K.S., D.J.B., M.M., B.H. and J.A.H.). The research was supported by the Wellcome Trust Core Award Grant Number 203141/Z/16/Z with additional support from the NIHR Oxford BRC. The views expressed are those of the author(s) and not necessarily those of the NHS, the NIHR or the Department of Health. We are grateful to all health workers and patients who supported collection of field samples in Zambia. We thank Kirk Rockett for providing laboratory space and support during initial stages of the project; staff at the Oxford Genomics Centre for sequencing support, especially Amy Trebes and David Buck; Nada Kubikova for discussions about multiplex PCR primer design; Gavin Band and Annie Forster for beta testing of *multiply*; Shazia Ruybal for discussions about *msp2* analysis. We also acknowledge the support of the Royal Geographical Society with IBG and Jaguar Land Rover for instigating initial collaborations through the 2018 RGS Land Rover Bursary which was awarded to J.A.H., I.G. and G.B.B.

## Author contributions

M.dC.: Methodology, Investigation, Writing - Review & Editing. M.M.: Methodology, Investigation, Writing - Review & Editing. A.J.: Methodology, Investigation, Validation, Writing - Review & Editing. J.C.: Resources; C.D.: Project administration, Writing - Review & Editing. K.S.: Project administration, Funding acquisition, Writing - Review & Editing. B.M.: Investigation; K.G.: Investigation; R.V: Conceptualisation, Formal Analysis; C.N.: Resources; M.C.: Resources; S.P.: Resources; J.A.B.: Supervision, Reviewing & Editing. I.G.: Conceptualisation, Funding acquisition, Writing - Review & Editing. G.B.: Conceptualisation, Funding acquisition, Writing - Review & Editing. B.H.: Project administration, Funding acquisition. M.H.: Project administration, Resources. D.J.B.: Conceptualisation, Supervision, Project administration, Funding acquisition, Writing - Review & Editing. J.A.H.: Conceptualisation, Supervision, Funding acquisition, Investigation, Formal analysis, Software, Visualization, Writing - Original Draft.

## Funding

## Competing interests

The authors declare no competing interests.
