## [Peer Review File · Nature Communications]

Flexible and cost-effective genomic surveillance of *P. falciparum* malaria with targeted nanopore sequencingReviewers' Comments:

Reviewer #1:

Remarks to the Author:

Mariateresa de Cesare and colleagues present a new protocol for MinION sequencing of up to 18 genomic markers for surveillance of drug resistance, *hrp2/3* deletion, and recrudescence typing. A strength of the manuscript is the development of a new software pipeline (called 'multiply') for the design of primers for multiplexed reactions. The main weakness is the overall limited validation on a small number of samples. Key criteria to judge the protocol, such as the ability to detect minority clones, limit of detection, usefulness for *hrp2/3* deletion typing are not or only partially investigated. The manuscript would benefit from additional samples being sequenced in order to address these issues, and a more detailed presentation of some of the current results.

Major comments

1) The development of the multiply protocol for primer design addresses a need of many researchers. Based on the manuscript, the success rate when using this protocol was 100%, all primers worked in the first try. It would be interesting to know how generalizable this finding is. Did the authors design any other primers, and did they always work?

Also, do 'off-target binding sites' considered by multiply refer to off-target binding within the *Pf* genome, or does it refer to the human genome? Does multiply screen against other blood-borne pathogens, e.g. *P. vivax*?

2) I struggle to understand why amplicons of 3-4 kb were designed, but mostly only the much shorter CDS were analyzed. The *msp2* amplicon is 3-fold longer than the CDS, several others a twice as long. Is this because multiply did not find other primers, or is there another reason?

As a result, the section "Producing long-read data for policy-relevant *P. falciparum* genes" is somewhat confusing. Rather than focusing on the percentage of the CDS covered, why not focus on how many full-length amplicons are sequenced?

3) The panel includes *hrp2* and *hrp3*, as deletions of these genes result in false-negative rapid tests. The main question around these targets is not genetic diversity, but showing whether the gene is present or absent. To this aim, various PCR-based protocols are published. A previous study evaluated MinION sequencing for deletion typing, and faced problems of false-positive calls (reads mapping to *hrp2/3* in samples with deletions) (<https://pubmed.ncbi.nlm.nih.gov/36801925/>). No effort has been undertaken to validate the protocol presented in the current study for deletion typing. This is a major limitation that needs to be addressed.

4) Supplementary Table 2 includes samples on mock mixtures of 2-4 strains at different ratios (but always at 10,000 parasites/uL, which is a major limitation). The data of these mixtures are hardly presented but would give some insights into the ability to sequence minority clones. They should be included in the main manuscript. A figure showing expected vs. observed proportions of reads would be useful.

5) It is difficult to understand why the authors selected *msp2* for recrudescence typing (and COI). *Msp2* is a great marker when sized by capillary electrophoresis, but as the authors state, the many indels make analysis of sequencing data challenging. Multiple shorter, highly diverse (but only containing SNPs) amplicons are published and tested for recrudescence typing. Figure 5 shows a lot of overlap in read lengths of clones that should carry alleles of different sizes (e.g. CamI543Tand 3D7). Likewise, even if most reads cluster around the expected size, how could one know whether other reads are sequencing artifacts or a true minority clone? This distinction is crucial for recrudescence typing. The results leave the impression that this protocol cannot be used for recrudescence typing, more discussion on that is needed.

The authors acknowledge that using a single marker for COI estimates is a limitation. Why did they not include more markers?

6) Mixtures of Pf and human DNA: The authors describe that they mixed 0.25 ng Pf DNA (corresponding to 10,000 parasites) with 0.25 ng human DNA. 0.25 ng human DNA, however, only corresponds to approximately 80 genomes (i.e. 1% of the 8000 WBCs found per μL of blood). These samples thus do not represent anything similar to a human blood sample, which might affect the number of reads mapping to human, and other parameters. This needs clarification. Was DNA quantified independently in the lab of the researchers after dilutions were made, e.g. by qPCR? When ordering DNA, concentrations might not always be accurate.

7) In the section "Effect of parasitemia on sequencing performance" it would be helpful to have a figure and/or table. It is hard to judge the results based on a statement like "The coverage fold-difference across amplicons was higher at lower parasitemia values". Rather than stating that "partially masked the effect of parasitemia and increased variation in coverage", it would be better to exclude samples with deletions from this analysis. Likewise samples with *mdr1* CNVs should be excluded. Did the authors make an attempt to quantify the 16 field samples with missing data by qPCR?

Minor comments

Introduction: Caution needs to be taken to state that "deletions causing false negative rapid diagnostic test (RDT) results have been detected at high frequency in Ghana". A more recent much larger study did not find deletions (<https://journals.plos.org/plosone/article?id=10.1371/journal.pone.0238749>)

What is the rationale to include flanking genes for *hrp2/3*? Only deletions of *hrp2/3* are expected to affect RDT performance.

The authors use the not well-known acronym VT for 'vaccine target' a single time in Table 1. I would recommend not to use it.

Figure 1: I am not sure whether the short sequence below the blue box "Day 3" is meant to be there.

Typo in Supplementary Figure S6 legend: "SNP calling errors in *mdr1* are have two primary sources."

Reviewer #2:

Remarks to the Author:

This work is technically sound and I congratulate the authors on their commitment to open access research with the use of github, protocols.io and bioRxiv so that all materials are available freely to researchers. However I question whether this approach achieves the stated aim of being a field-deployable method. The laboratory protocol is time consuming, expensive and too dependent on specific reagents which will be difficult/impossible to buy in most African countries without support from outside laboratories.

Not much consideration seems to have been given as to how to make the protocol easier. The WGA amplification takes >16h which could be replaced with a 2-4h reaction using a thermostable mutant which would also improve the representation of high AT sequences. The PCR takes >4h which could have been sped up by moving to a two-step reaction as the annealing step is likely not functional at 15 nM primer concentration. The ligation barcoding takes >4h which could be replaced by a 0.5 h rapid barcoding protocol as the full-length reads don't appear necessary for the analysis performed.

Currently this is similar in time and cost to hybridisation capture methods so the technical explanation for this approach being preferred needs to be made.

It needs to be clearly explained why the WGA is used at all given it was developed as a method for untargeted Illumina sequencing. In this multiplex PCR application it should not be required unless the PCR itself generates a huge amount of off-target sequence. This may be the case but this needs to be explained because I think many people will be questioning why you need both WGA and PCR amplification to generate these amplicons for sequencing.

The justification for use of long amplicons needs to be made more clearly as there are many obvious disadvantages to this including lower sensitivity. Small amplicons would suffice to determine mutations involved in drug resistance and monitor diagnostic targets. Primers could be designed inside CDS with higher sequence complexity and T_m which would in turn improve specificity.

The analysis of SNP calling accuracy in low complexity sequences has illustrated the progress made from earlier attempts to use nanopore sequencing for Plasmodium surveillance. Clearly systematic errors still remain in low complexity intergenic regions but this work was done using an old chemistry version which has been replaced with the R10.4.1/LSK114 so the authors should flag this limitation of the study.

Reviewer #3:

Remarks to the Author:

The manuscript by de Cesare et al titled "Flexible and cost-effective genomic surveillance of *P. falciparum* malaria with targeted nanopore sequencing", explores the possibility of the nanopore sequencing for targeted sequencing of the representative genes which are related to the drug resistance of malaria parasites. The authors started with designing a panel of PCR primers for eight or sixteen key genes by using their own software of this purpose, named multiply. They PCR-amplified the target regions and used the obtained PCR products of 3-4kb in length for the nanopore sequencing. They evaluated the effectiveness of the amplification depending of varying parasitemia, firstly using in vitro culture parasites and then, clinical samples stored on the DNA-absorbing papers.

I totally agree with the authors idea that the nanopore sequencing has a great potential to change the current way to conduct the sequencing-based genotyping. However, I have to point out that the scientific advances described in this particular paper is quite limited. After all, the originality of this paper lies in the application of their novel PCR design program for the malaria genome. This issue is practically very important, but there has been a large number of programs developed for this purpose. And it is not easy to precisely evaluate their performance, as the PCR amplification is heavily dependent on the sample conditions, PCR reagents, target length/GC contents and skill of the technicians.

Major points:

1. Since the initial application of the nanopore sequencer for the PCR-amplicon sequencing of Ebola virus (Quick et al Nature 2016), there has been a long history of the sequenced-based genome typing analysis for various pathogens on the nanopore platform. Several papers have also reported the analysis of drug resistant genes in malaria parasite (Runtuwene et al, Sci Rep 2018). In my opinion, in spite of those methodological developments, the "on-site" sequencing has not yet become the "standard" approach. There are several reasons, but the largest one may be that the sample preparation (DNA extraction, WGA and PCR amplification; above all the library preparation for the sequencing) is not easy to be completed in a field condition in most of the cases. For the research purpose, practically, the samples are transferred to a "central" lab at some stage. But, once they are transferred to the laboratory, the short read sequencing (of shotgun amplicons) is the easier and more cost-effective method. On the medical practice base, I wonder if the real-time PCR should be the first choice, at the first place, once the key mutations have been identified. I'm not convinced the authors have successfully addressed these issues in this particular paper.

2. Long read nature of the nanopore sequencing may provide a different potential from the other short read platforms. However, the amplicon size presented here is not particularly long, and setting long targets would rapidly decrease the success rate of the amplification. Also, having a haploid genome at the blood stage, the advantage in using the long read sequencing for the malaria genotyping may be limited rather than other pathogens having diploid genomes or having a high rate of co-infections.

3. Rather than the PCR-based selection of the target regions, making use of the "adaptive sampling" function of nanopore sequencing should be, at least, easier. Have the authors tried this approach?

Minor points:

4. With the advent of the new flow cell (R10.4 and kit 14), I think it may be timely for additional analysis regarding Figure 4b, to see whether the new version is appropriate to handle homopolymer regions. For example, it has been known that MinION performance in AT-rich and homopolymer (as in the introns of pfCRT) regions are very erroneous, but its performance in the CDS is reliable.

5. Eventual success rates of the diagnosis (on the absence or presence of the drug resistant mutations) should be presented in a more explicit manner, depending on the varying parasitemia and condition of the samples.

Miscellaneous points:

6. Please mention that DBS is the abbreviation of dry blood spot in the beginning of the manuscript.

7. The multiply software seems to require VCF file. If we are to create primers of unknown sequences, then VCF file may not be available. How do handle it in this situation? Or is VCF file is optional?

8. On page 7, it is written that 57.5% of total sequencing were on-target. But Figure 2a depicts 57.4%.

9. On page 8, section "Effect of parasitemia on sequencing performance", the authors evaluated the metrics for four sequencing experiments described before. Which four experiments did the authors refer to?

10. Also, in the same section, the authors mentioned that they did not perform any sample normalization, but y-axis of Figure 3 top panel depicts normalized sample throughput.

11. On page 10, the authors mentioned that 150 in-silico replicates, but the sample size for this experiment is 30. Please clarify.

12. Does Clair3 software create a consensus sequence of the reads before calling the SNPs?

Responses are written in blue

We thank all three reviewers for their insightful comments.

In response, we have added three new major sequencing experiments to the manuscript. First, to address concerns around deployability in Africa, and to also provide additional validation, we include analysis for 41 samples prepared and sequenced entirely at a government laboratory at the National Malaria Elimination Centre in Zambia. Second, to confirm the ability of our assay to detect *hrp2/3* deletions, we conducted a proof-of-principle experiment using 48 mock samples with a range of parasitemia levels; and also developed a novel statistical model facilitating rigorous inference of these deletions. Third, we repeated our characterisation of *P. f.* SNP calling accuracy using the latest flow cells (R10.4.1) and chemistry (LSK114), along with the highest accuracy basecalling algorithm (dorado v0.34), basecalling model (SUP), and variant caller (Clair3 v1.04) – providing a state-of-the-art picture of ONT performance for the unique and important use case of malaria genomic surveillance. In addition, we explored some of the protocol optimisations suggested by Reviewer #2; share some unpublished data for another multiplex developed using MULTIPLY; and demonstrate the benefit of sWGA in our protocol. Lastly, we have made textual changes to the manuscript to clarify results and further discuss the challenges and opportunities associated with field-deployable sequencing.

We acknowledge that Reviewer #1 made additional experimental requests, in particular around recrudescence typing and minor clone detection in polyclonal infections. Although such experiments are of interest to us, to conduct analyses in a meaningful way would require the development of novel bioinformatics tools (i.e. haplotype inference methods) that can handle the longer and higher error nanopore reads, in the challenging context of samples with an unknown ploidy and unknown haplotype proportions. While such methods exist for short-read Illumina data (e.g. DADA2, SeekDeep, &c), no methods currently exist for ONT data. Indeed, their development, benchmarking and careful validation represents a major statistical, computational and experimental undertaking in the field. We would also like to highlight that we comment on this as a limitation in the Discussion.

Here, we have developed novel, open-source software for design of primers for multiplex PCR; created the two largest publicly available long-read amplicon panels for *P.f.*, which target critical drug, diagnostic, and vaccine targets; reduced ONT amplicon sequencing costs for *P.f.* to USD \$25 / sample through novel protocol optimisations; validated the assay on a suite of mock and field samples, and deployed our approach in Africa. We feel that also producing a novel bioinformatic method for haplotype inference is outside the scope of this manuscript.

More detailed responses to specific reviewer comments are included below.

Reviewer #1 (Remarks to the Author):

Mariateresa de Cesare and colleagues present a new protocol for MinION sequencing of up to 18 genomic markers for surveillance of drug resistance, *hrp2/3* deletion, and recrudescence typing. A strength of the manuscript is the development of a new software pipeline (called 'multiply') for the design of primers for multiplexed reactions. The main weakness is the overall limited validation on a small number of samples. Key criteria to judge the protocol, such as the ability to detect minority clones, limit of detection, usefulness for *hrp2/3* deletion typing are not or only partially investigated. The manuscript would benefit from additional samples being sequenced in order to address these issues, and a more detailed presentation of some of the current results.

Thank you for your time and detailed feedback on our work.

We conducted validation using a panel of 24 mock samples and 28 field samples, which we feel is informative and also comparable to several other manuscripts in this area. In terms of evaluation criteria, in a first instance we tried to focus on the most salient features of the assay – namely the coverage generated (Fig 2), its performance across parasitemia levels (Fig 3), and its ability to accurately call known drug resistance mutations (Fig 4).

We agree, however, that more validation would strengthen the manuscript. To that end, we include an additional 41 field samples that were sequenced entirely in a container lab at the National Malaria Elimination Centre in Zambia. Moreover, we include an experiment with mock samples to investigate assay performance at *hrp2/3* deletion typing and across parasitemia levels.

Validation of features related to polyclonal *P. falciparum* infections (such as minor clone detection) are of significant interest, but at present we limit our exploration to what is presented in Fig 5. The main reason for this is that to properly investigate these features would require a way to infer individual haplotypes (or call genotypes) within polyclonal infections – and no such methods currently exist that can handle nanopore data. Development of such methods is now underway using the data generated from our assays, but as highlighted above we feel their inclusion is beyond the scope of this paper. This point is highlighted in paragraph 3 of the discussion as a limitation and an important future direction.

Major comments

1) The development of the multiply protocol for primer design addresses a need of many researchers. Based on the manuscript, the success rate when using this protocol was 100%, all primers worked in the first try. It would be interesting to know how generalizable this finding is. Did the authors design any other primers, and did they always work?

Also, do 'off-target binding sites' considered by multiply refer to off-target binding within the Pf genome, or does it refer to the human genome? Does multiply screen against other blood-borne pathogens, e.g. *P. vivax*?

During the project we did try two different amplicon panel designs for the same targets. The first was a panel where amplicon lengths varied from 2-6kbp (fully spanning the CDS of all of our targets), and the second where lengths were restricted to 3-4kbp (where we had to truncate *mdr1*). Both of these primer sets looked to be working in preliminary experiments (see the figure below and Supplementary Fig 2). However, the panel with more variation in amplicon lengths appeared to have worse balance across the amplicons, hence the decision to proceed with the 3-4kbp panel.

Figure 1. Validation of 2-6kbp multiplex PCR (internally called MA01b) for NOMADS8 targets. Validation approach is the same as displayed in Supp. Fig 1 of the manuscript. Note how the multiplex PCR produces products of the expected size for every target, but *mdr1* (5.6kbp) has low abundance, and *pml* (2.0kbp) has high abundance, as assessed by agarose gel. In the absence of the multiplex PCR, no individual target PCR products are observed at an equivalent dilution of the starting material.

This of course does not imply that the success rate of the software is 100%, nor do we claim this in the manuscript. As highlighted by Reviewer #3 at the end of their introductory statement, it is very difficult to give an estimate for how well multiplex primer design software will perform that is generalisable. This is because success rates are strongly dependent on the sample type, the complexity of the panel one is trying to design (e.g. number of targets, their length, the specific sequences being amplified), reagent selections and other technical factors. Systematically exploring these factors in a statistically meaningful way would be a leviathan task. However, we agree that it is interesting and as we, and hopefully the wider scientific community, use the software more, we plan to keep track of what settings are associated with success and where more iteration is necessary.

For the version of MULTIPLY used to design the panels in the manuscript, we only screen for off-target binding sites within the *P.f.* genome. There were two reasons for this: (i) at the time we were already convinced that we should include sWGA to increase the fraction of template that was *P.f.*; and (ii) blasting several hundred candidate primers against *H.s.* is rather slow computationally.

We agree that screening against other genomes does make sense for many applications and we hope to include it as an option for the user in future software releases.

2) I struggle to understand why amplicons of 3-4 kb were designed, but mostly only the much shorter CDS were analyzed. The *msp2* amplicon is 3-fold longer than the CDS, several others a twice as long. Is this because multiply did not find other primers, or is there another reason?

As a result, the section "Producing long-read data for policy-relevant *P. falciparum* genes" is somewhat confusing. Rather than focusing on the percentage of the CDS covered, why not focus on how many full-length amplicons are sequenced?

There were two reasons we chose to develop a panel with 3-4kbp amplicons, despite this meaning some amplicons are considerably larger than their target gene's CDS.

The first, and as we alluded to above, is to avoid amplification bias towards shorter targets. Such bias would cause greater imbalance in coverage across our targets and reduce the number of samples that could be sequenced per flow cell. Note that bias towards shorter targets is very well documented (for example, see Messerli *et al.* (2017) Antimicrobial Agents and Chemotherapy), was visible already in preliminary experiments with the 2-6kbp panel, and makes mechanistic sense as extension efficiency is a decreasing function of amplicon

length. Therefore, given that several of our target genes have a CDS >3kbp, we designed a panel with 3-4kbp amplicons (by setting the minimum and maximum amplicon length accordingly in MULTIPLY).

A second point is that we have a longer-term belief that as the ONT error rates continue to drop, and suitable bioinformatic methods are developed, there will be valuable information on genetic diversity in intergenic regions; for example, in short-tandem repeats. We are looking forward to exploring this in the coming years.

At present, we focussed primarily on the CDS because this is the critical region for detecting drug-resistance associated mutations, and also as we demonstrate in Fig 4, is possible with high accuracy given current error rates and bioinformatic methods.

3) The panel includes *hrp2* and *hrp3*, as deletions of these genes result in false-negative rapid tests. The main question around these targets is not genetic diversity, but showing whether the gene is present or absent. To this aim, various PCR-based protocols are published. A previous study evaluated MinION sequencing for deletion typing, and faced problems of false-positive calls (reads mapping to *hrp2/3* in samples with deletions) (<https://pubmed.ncbi.nlm.nih.gov/36801925/>). No effort has been undertaken to validate the protocol presented in the current study for deletion typing. This is a major limitation that needs to be addressed.

Thank you for bringing the *Sabin et al.* reference to our attention – we had missed it and it is now cited in the manuscript.

We have conducted a validation experiment exploring the ability of our assay to investigate *hrp2/3* deletions across a panel of clonal mock samples with varying parasitemia. In the experiment and subsequent analysis we demonstrate that we are able to clearly detect both *hrp2* and *hrp3* deletions. Although we do observe a handful of reads mapping to *hrp2/3* in samples where they should be deleted, these reads were between ten- and a hundred-fold less abundant than in *hrp2/3* positive samples.

We note that our approach has several helpful advantages compared to *Sabin et al.*. In particular, the use of longer reads will likely have reduced mapping errors; the inclusion of flanking targets helps to strengthen the evidence in favour of, or against, a deletion; and the other targets within our multiplex can provide contextual information about the quality of the sample and presence of amplifiable DNA. All of this combined greatly minimises issues with false-positive (or negative) calls. As a proof-of-principle, we developed a novel statistical model that we use to make rigorous *hrp2/3* deletion calls from our data, taking into account read misclassification and variation in sample quality. With this model we are able to show that all expected deletions are identified with a very high degree of confidence.

4) Supplementary Table 2 includes samples on mock mixtures of 2-4 strains at different ratios (but always at 10,000 parasites/uL, which is a major limitation). The data of these mixtures are hardly presented but would give some insights into the ability to sequence minority clones. They should be included in the main manuscript. A figure showing expected vs. observed proportions of reads would be useful.

We feel that we have presented these mixtures to a suitable degree; the signals of polyclonality in these mixtures are explored using three different approaches and presented in Figure 5 and the associated results section. Figure 5a, for example, contains information about both the expected and observed proportions of reads, in the sample names and corresponding points in the scatterplot.

Second, note that in the results section we also describe a limitation of our experimental setup, namely that the laboratory strain DNA we procured from BEI resources seems to be of variable quality. As a result, some strains (Dd2, HB3) systematically sequenced better than others, and this means the observed read proportions are biased in a way that: (i) would not occur in samples collected in the field; and (ii) cannot be meaningfully linked to a “true” minority clone proportion. As such, these mixtures, while useful for exploring ways to detect and characterise polyclonal infections, are not appropriate for evaluating the limits of minor clone detection.

Last and perhaps most importantly, we again emphasise that meaningful analysis around the detection of minor clones is beyond the scope of this manuscript. How accurately minority clones are ultimately detected will depend not just on the raw data, but critically on *how* a bioinformatic method uses all of the information within that data, and potentially integrates it with salient prior information (viz. Bayesian approaches). To do this properly, novel

bioinformatic methods will need to be developed that can handle the long and noisy reads generated by nanopore sequencing, in the context of the difficult case of unknown ploidy and proportions.

5) It is difficult to understand why the authors selected *msp2* for recrudescence typing (and COI). *Msp2* is a great marker when sized by capillary electrophoresis, but as the authors state, the many indels make analysis of sequencing data challenging. Multiple shorter, highly diverse (but only containing SNPs) amplicons are published and tested for recrudescence typing. Figure 5 shows a lot of overlap in read lengths of clones that should carry alleles of different sizes (e.g. Cam1543T and 3D7). Likewise, even if most reads cluster around the expected size, how could one know whether other reads are sequencing artifacts or a true minority clone? This distinction is crucial for recrudescence typing. The results leave the impression that this protocol cannot be used for recrudescence typing, more discussion on that is needed.

The authors acknowledge that using a single marker for COI estimates is a limitation. Why did they not include more markers?

There are two separate issues of relevance when it comes to marker selection for recrudescence typing and COI estimation: (i) the local genetic variation of the marker(s) selected and (ii) the ability of existing methods to discriminate that variation. As you acknowledge *msp2* is a great marker by CE, we understand your concern to be limited to the second point: discrimination using sequencing data.

Certainly, indels (or more precisely for *msp2*, tandem repeat variation) can be more challenging to infer accurately from sequencing data than SNPs. However, this primarily applies when analysis is done with a naïve reference-based approach (i.e. when the sample diverges dramatically from the reference used), and/or when the length of the variable region is similar or greater than the length of the reads. In both cases, mapping and/or alignment can be ambiguous and error-prone. However, neither of these cases apply to our analysis or data.

First, regarding the analysis, we are quite explicitly against taking a naïve-reference based approach to *msp2* in the manuscript. To this end, we exclude *msp2* from the analysis in Fig 4 and show three alternative approaches in Fig 5: (i) length polymorphism, (ii) mapping to multiple references, and (iii) pairwise alignment / clustering of reads.

Second, regarding the read lengths, our amplicon is 3.7kbp which is ~7X longer than the ~500bp central variable region of *msp2*. Short-read approaches typically use amplicons of ~200-500bp and generate reads of ~150-300bp; ultimately, they are not well-suited to grapple with the variation in *msp2*. This is part of what motivated various teams to look towards the other high diversity genes you allude to – like *ama1*, *cpmp*, *csp*, *sera2*, and *trap* – despite *msp2* having excellent population-genetic properties for COI estimation or recrudescence typing. What we were excited to convey with Fig 5 was that with long-read amplicon sequencing, these excellent population-genetic properties of *msp2* can be fruitfully harnessed.

We acknowledge there is overlap in read lengths between Cam153T and 3D7 – but point out that the two distributions are easily distinguishable by the basic approach of aligning to just four different reference alleles (this can be seen by the uniformly distinct colours of the points in the two distributions in Fig 5a). Although we showed that read-lengths contain some informative signal, we would not advocate for only using them – hence they were just one of three different analyses displayed in Fig 5. We agree that only using the length of reads (i.e. ignoring the sequence polymorphism itself) would be an unjustified and inferior way to conduct this analysis.

Determining whether faint signal is a technical artefact or a minority clone is a challenge for all recrudescence typing methods. Just like with other methods, an approach using long-read *msp2* sequencing data could either handle this probabilistically or with an informed threshold. Ultimately, there will be a limit of detection that would be a function of sequencing depth.

Finally, we do include two other highly diverse markers (*ama1* and *csp*) in our NOMADS16 panel that will benefit recrudescence typing. While it would be great to add even more markers, this needs to be balanced against the complexity introduced to the resulting multiplex PCR (e.g. the number of primer-primer interactions grows exponentially in the number of targets). With 16 targets amplifying ~55kbp we already have the largest existing long-read amplicon panel for *P. falciparum* by an order of magnitude. As additional bioinformatic and statistical

methods are developed, we are confident a panel of this size will already support high quality recrudescence typing.

6) Mixtures of Pf and human DNA: The authors describe that they mixed 0.25 ng Pf DNA (corresponding to 10,000 parasites) with 0.25 ng human DNA. 0.25 ng human DNA, however, only corresponds to approximately 80 genomes (i.e. 1% of the 8000 WBCs found per uL of blood). These samples thus do not represent anything similar to a human blood sample, which might affect the number of reads mapping to human, and other parameters. This needs clarification.

Was DNA quantified independently in the lab of the researchers after dilutions were made, e.g. by qPCR? When ordering DNA, concentrations might not always be accurate.

Thank you for bringing this point to our attention – this was a typo in the manuscript. We diluted the 0.25ng/uL of *P.f.* DNA in **25ng/uL of human DNA**; which gives approximately ~10,000p/uL. This is then diluted to ~1ng/uL before going to sWGA, which is a typical concentration we get from Qiagen extractions of DBS in the field.

We are confident that our mock samples are a reasonable recapitulation of what can be observed in the field. For example, they performed similarly to the field samples in Fig 2.

7) In the section “Effect of parasitemia on sequencing performance” it would be helpful to have a figure and/or table. It is hard to judge the results based on a statement like “The coverage fold-difference across amplicons was higher at lower parasitemia values”.

Rather than stating that “partially masked the effect of parasitemia and increased variation in coverage”, it would be better to exclude samples with deletions from this analysis. Likewise samples with *mdr1* CNVs should be excluded.

Did the authors make an attempt to quantify the 16 field samples with missing data by qPCR?

Fig 3 of the manuscript presents all of the data in this section; for example, your specific question about coverage is shown in the bottom panel, where there are negative correlations in the scatterplots. Regarding excluding samples with *hrp2/3* deletions or *mdr1* amplifications, although we understand your point of view, we feel this is a subjective matter. In our opinion, it is more informative to include as many samples as possible in the analysis – especially if similar types of samples will occur in nature – and then show in isolation the effect that may be caused by unusual genotypes. In particular we think this approach is preferable when two of the major lab strains (Dd2 and HB3) both contain *hrp2/3* deletions and potentially *mdr1* duplications.

As regards the missing qPCR data, unfortunately the original extracted genomic DNA was exhausted so we could not quantify parasitemia by qPCR. Happily, the additional set of field samples collected as part of this review all have parasitemia data.

Minor comments

Introduction: Caution needs to be taken to state that “deletions causing false negative rapid diagnostic test (RDT) results have been detected at high frequency in Ghana”. A more recent much larger study did not find deletions (<https://journals.plos.org/plosone/article?id=10.1371/journal.pone.0238749>)

Thank you. We missed this reference and have updated the list of countries where deletions have been detected.

What is the rationale to include flanking genes for *hrp2/3*? Only deletions of *hrp2/3* are expected to affect RDT performance.

The deletions of *hrp2* and *hrp3* that have been observed to date are large and also result in the loss of flanking genes. This was true of the original deletions that were discovered in Peru in 2010 (<https://journals.plos.org/plosone/article?id=10.1371/journal.pone.0008091>) and of the deletions now prevalent in Ethiopia (<https://www.nature.com/articles/s41564-021-00962-4>), Eritrea (https://wwwnc.cdc.gov/eid/article/24/3/17-1723_article), and other parts of Africa and Southeast Asia

(<https://wellcomeopenresearch.org/articles/8-22>). Including PCR targets involving the flanking genes is relatively common practice for *hrp2/3* deletion detection and increases the strength of the evidence.

In particular when designing NOMADS16, we were influenced by a review on best practices for *hrp2/3* deletion detection (<https://malariajournal.biomedcentral.com/articles/10.1186/1475-2875-13-283>), in which the authors make three recommendations for PCR assays: (1) perform a PCR of full length exon 2 as well as the region bridging exon 1 and 2, to allow for both entire and partial gene deletions; (2) perform a PCR amplifying at least two other single copy genes; (3) perform a PCR of flanking genes. NOMADS16 satisfies these recommendations in a single multiplex PCR.

We have added a paragraph to the discussion around *hrp2/3* deletion detection and mention these points.

The authors use the not well-known acronym VT for 'vaccine target' a single time in Table 1. I would recommend not to use it.

We have removed the acronym.

Figure 1: I am not sure whether the short sequence below the blue box "Day 3" is meant to be there.

It was there deliberately, but we see how it could be confusing and have removed it.

Typo in Supplementary Figure S6 legend: "SNP calling errors in *mdr1* are have two primary sources."

Thank you for bringing this to our attention, it has been corrected.

Reviewer #2 (Remarks to the Author):

This work is technically sound and I congratulate the authors on their commitment to open access research with the use of github, [protocols.io](https://www.protocols.io) and bioRxiv so that all materials are available freely to researchers. However I question whether this approach achieves the stated aim of being a field-deployable method. The laboratory protocol is time consuming, expensive and too dependent on specific reagents which will be difficult/impossible to buy in most African countries without support from outside laboratories.

Thank you for your thoughtful review.

To address some of your concerns, it would be useful to establish a common understanding of the term “field-deployable” (or “deployable in a low-resource setting”, which is our phrasing in the manuscript). In our view, this terminology requires that an assay uses only widely available or low-cost equipment that can be procured and maintained in a basic laboratory facility (i.e. those typically found across malaria-endemic Africa). Second, the reagents required should be deliverable and easily storable. Third, the protocol shouldn’t be so complicated as to make training prohibitive or data generation unreliable. Fourth, for widespread use, the per-sample costs need to be reasonable. We are confident our assay satisfies all these conditions.

In support of this view, we note that during the pandemic nearly 100,000 SARS-CoV-2 genomes were sequenced in Africa (<https://www.science.org/doi/10.1126/science.abq5358>). Over half of these used the MinION and generated data from local labs (our team generated ~2,000 SARS-CoV-2 genomes in Zambia this way). For labs deploying the ARTIC protocol for nanopore sequencing of SARS-CoV-2, the transition to our protocol for *P. falciparum* is relatively straightforward. Critically, the equipment, reagents, protocol complexity, and per-sample costs are all similar.

We agree that procurement and supply chain issues are more common in Africa, but this is not a unique feature of our assay – it affects all types of molecular work done there. Moreover, in our experience the situation is improving, with a growing list of suppliers allowing local teams to procure (e.g. Carramore, Whitehead Scientific, LASEC, &c) combined with stronger support and commitment from actors like ONT, Africa CDC and BMGF. We don’t think these global complexities imply that our assay is not field-deployable.

Additionally, we would like to highlight that the new field samples included in response to this review were prepared and sequenced in a government facility container lab at the National Malaria Elimination centre (NMEC) in Zambia, by our local team members. Ultimately, we are confident that our protocol is field-deployable – both because similar protocols have been used to sequence thousands of viral genomes across Africa, and because we are currently using it.

Not much consideration seems to have been given as to how to make the protocol easier. The WGA amplification takes >16h which could be replaced with a 2-4h reaction using a thermostable mutant which would also improve the representation of high AT sequences. The PCR takes >4h which could have been sped up by moving to a two-step reaction as the annealing step is likely not functional at 15 nM primer concentration. The ligation barcoding takes >4h which could be replaced by a 0.5 h rapid barcoding protocol as the full-length reads don’t appear necessary for the analysis performed. Currently this is similar in time and cost to hybridisation capture methods so the technical explanation for this approach being preferred needs to be made.

Thank you for your insightful suggestions around potential improvements to our protocol.

During protocol development our team eliminated all but one of the SPRI clean-up steps, which are the most technically challenging and error-prone in most NGS workflows. We also adapted a “one-pot” strategy to simplify pipetting during the barcoding process (components are added consecutively to a single 96-well plate, without plate transfers or sample normalisation required); on top of exploring volume reductions in the sWGA and barcoding procedure. Our objectives with these optimisations were (i) to simplify the hands-on process as much as possible and (ii) to reduce costs. At the same time, we agree with you that reducing incubation times and speeding up the protocol is another worthwhile optimisation objective and we have explored some of your suggestions, as detailed below.

We believe by a “thermostable mutant” you are referring to EquiPhi29. If so, we were unaware Phi29 had been optimised and we thank you for bringing this to our attention. In response, we conducted several trial experiments where we substituted Phi29 with EquiPhi29 in our protocol (Figure 2). Very encouragingly we were able to generate robust PCR products from our NOMADS8/NOMADS16 panels after only 3hr of amplification with EquiPhi29 (Figure 2a). Surprisingly, however, we were unable to generate any sequencing reads for these PCR products (Figure 2b), despite generating reads for matched Phi29 samples in the same library prep, ruling out any procedural errors.

Figure 2. Exploring substituting Phi29 with EquiPhi29. (a) Agarose gel results for a set of three mock samples where EquiPhi29 was used instead of Phi29 for sWGA. EquiPhi29 was tested both with- and without pre-denaturation of the template DNA. Robustness to cold-chain disruption was explored by leaving an aliquot of EquiPhi29 at RT for 1 or 2 days. Top panel shows NOMADS8 PCR products, and bottom shows NOMADS16 PCR products. The same mock samples were used to generate data using the standard Phi29 approach

described in the manuscript. (b) Number of reads generated for a subset of the samples in (a) after following the NOMADAS ligation protocol and sequencing on a R10.4.1. Flow Cell using a MinION Mk1b device. Reads were only generated for samples where Phi29 was the enzyme in sWGA.

In spite of the intervening SPRI step, we hypothesised that the most likely cause was inhibition of the barcoding ligation reaction in the EquiPhi29 samples. We tried to reduce the exposure of the ligation reaction to any sWGA components or byproducts by: (i) reducing the EquiPhi29 reaction time (trying 0.5hr, 1hr, and 2hr); (ii) by serial diluting the sWGA product before using it as template in the PCR reaction (up to 8-fold); and (iii) by adding less PCR product into the ligation reaction. Even at the lowest EquiPhi29 run times and highest dilutions, we still observed robust PCR product; however, yet again, we could not generate any sequencing reads (data not shown).

We have contacted a ThermoFisher representative and they have confirmed that the EquiPhi29 buffer does contain something that would inhibit ligation, but felt it should be removed by the SPRI step. In summary, we will continue to explore incorporating EquiPhi29 into our protocol (we are considering trying two-sided SPRI cleans, and different ligases) because we agree it would be great if we could so dramatically reduce the sWGA time. However at present, we are unable to get this to work and will stick with Phi29 which is reliable and is backed by a published protocol for *P.f.*. We welcome any troubleshooting suggestions you have for EquiPhi29.

Regarding the multiplex PCR, the majority of the runtime is spent in the 6min extension step (3hrs or ~86%) – the annealing step is only 15s per cycle (~8mins total, perhaps a little more including ramping time). This extension time boosts the efficiency of long-amplicons, especially in the context of the reduced extension temperature. Furthermore, longer extension times have been shown to reduce the rate of chimera artefacts in PCR, so we feel it is better to maintain it. However, we did conduct a preliminary experiment exploring whether the annealing step could be removed in response to your suggestion, and it looks as though it can (Figure 3). Given that the time savings here are very modest (especially compared to what we may get from EquiPhi29), we would like to conduct further validation to empirically verify amplicon balance or PCR robustness is not affected by this change. If this is the case, we will change to a two-step multiplex PCR in our next protocol update on protocols.io. As an aside we will also likely switch to the HotStart ReadyMix [KK2601] version of the KAPA kit, which makes setup a bit easier.

Figure 3. Comparing three-step and two-step multiplex PCR for NOMADS8. Agarose gel results for the NOMADS8 multiplex PCR for a 10,000p/uL mock sample. In the left most lane, the current multiplex PCR program including the annealing step is shown. From the right of that lane onwards, results for a two-step PCR

prepared from the same mastermix are shown across an extension temperature gradient. PCR was done using KAPA HiFi HotStart ReadyMix Kit.

Lastly, we have previously explored using the Rapid Barcoding Kit from ONT, despite it being recommended for genomic DNA input rather than amplicons. Although it is easier, in our experience yield from rapid-barcoded samples on the MinION was substantially lower than those prepared with ligation-based barcoding. Furthermore, note that we do perform some long-read analysis in Fig 5 with *msp2* – the fact that we don't use the full-length data more is not because it is not valuable but because appropriate bioinformatic methods are still being developed.

It needs to be clearly explained why the WGA is used at all given it was developed as a method for untargeted Illumina sequencing. In this multiplex PCR application it should not be required unless the PCR itself generates a huge amount of off-target sequence. This may be the case but this needs to be explained because I think many people will be questioning why you need both WGA and PCR amplification to generate these amplicons for sequencing.

Although it is true that sWGA was originally developed as a method for whole-genome sequencing on Illumina platforms, it has since been used in other applications including as a way to increase the amount of target template material prior to PCR. As two relevant examples, consider that MalariaGEN (who developed sWGA) use it as a preamplification step prior to amplicon sequencing on Illumina (<https://www.malariagen.net/resources/amplicon-sequencing-toolkit/p-falciparum-amplicontoolkit-protocols>) as does a team from Broad Institute (<https://onlinelibrary.wiley.com/doi/10.1111/1755-0998.13622>). With a spin-column extraction from a DBS, it is typical to get 0.5 - 1ng / uL of genomic DNA. Even at a relatively high parasitemia of 10,000p/uL, this will be 99% human, so you will have 0.005-0.01ng/uL parasite DNA. Given this starting point, it does make sense that sWGA would improve PCR performance. We now mention these points more explicitly in the discussion.

Personally, early in the project we did experiments and found that sWGA improved amplification for longer amplicons (e.g. >2kbp) using mock infections (Figure 4). We have not evaluated whether this is because it increases sensitivity (i.e. absolute amount of template), specificity (i.e. ratio of on- vs. off-target), or both.

Regardless, we are confident that for our amplicon panels, it significantly improves performance; and with our optimisations contributes only marginally to total cost.

Figure 4. sWGA increases the robustness of PCR for a variety of singleplex amplicons target drug resistance genes. Eight primer sets targeting different antimalarial drug resistance genes were tested in five contexts: (i) A mock mixture of 20% *P.f.* / 80% *H.s.* gDNA at 1ng/uL, as a “post-sWGA” positive control; post-sWGA template from a 1% *P.f.* / 99% *H.s.* gDNA mixture, either with a SPRI (ii) or without a SPRI (iii); a 1% *P.f.* / 99% *H.s.* gDNA mixture with a conventional PCR (iv) or a touch-down PCR (v) representing pre-sWGA samples. Note GB4 is a *P. falciparum* labstrain.

The justification for use of long amplicons needs to be made more clearly as there are many obvious disadvantages to this including lower sensitivity. Small amplicons would suffice to determine mutations involved in drug resistance and monitor diagnostic targets. Primers could be designed inside CDS with higher sequence complexity and Tm which would in turn improve specificity.

We spend the second last paragraph of the discussion outlining the benefits of long-read sequencing for malaria genomic surveillance, but we will reiterate some critical points here. Short-read panels are impelled to focus on ~250bp around known, common mutations. As a consequence, they can miss important emerging mutations. There are many pertinent and recent examples of this, including the P441L, C469Y and A675V mutations in *kelch13* that are emerging in Africa (most short-read panels were built around the C580Y mutation seen in SE Asia); as well as the piperazine associated mutations in *crt* which are outside most panels focussed on K76T (e.g. F145I, M343L, G343V); and there are now preliminary findings that N500E in *mdr1* is linked to lumefantrine resistance, yet most panels short-read panels focus exclusively on N86Y. Beyond this, important targets like *csp* are hard to explore with short-reads due to their repetitive regions; and the phenotype of some targets like *dhfr* and *dhps* depend on multiple mutations too far apart to be captured by a single short-read amplicon.

The analysis of SNP calling accuracy in low complexity sequences has illustrated the progress made from earlier attempts to use nanopore sequencing for Plasmodium surveillance. Clearly systematic errors still remain in low complexity intergenic regions but this work was done using an old chemistry version which has been replaced with the R10.4.1/LSK114 so the authors should flag this limitation of the study.

We have updated this analysis using R10.4.1/LSK114 chemistry, as well as the most current software.

Reviewer #3 (Remarks to the Author):

The manuscript by de Cesare et al titled “Flexible and cost-effective genomic surveillance of *P. falciparum* malaria with targeted nanopore sequencing”, explores the possibility of the nanopore sequencing for targeted sequencing of the representative genes which are related to the drug resistance of malaria parasites. The authors started with designing a panel of PCR primers for eight or sixteen key genes by using their own software of this purpose, named multiply. They PCR-amplified the target regions and used the obtained PCR products of 3-4kb in length for the nanopore sequencing. They evaluated the effectiveness of the amplification depending of varying parasitemia, firstly using in vitro culture parasites and then, clinical samples stored on the DNA-absorbing papers.

I totally agree with the authors idea that the nanopore sequencing has a great potential to change the current way to conduct the sequencing-based genotyping. However, I have to point out that the scientific advances described in this particular paper is quite limited. After all, the originality of this paper lies in the application of their novel PCR design program for the malaria genome. This issue is practically very important, but there has been a large number of programs developed for this purpose. And it is not easy to precisely evaluate their performance, as the PCR amplification is heavily dependent on the sample conditions, PCR reagents, target length/GC contents and skill of the technicians.

Thank you for your thorough and thoughtful review of our manuscript.

Although there is precedent for nanopore sequencing of *P. falciparum* malaria – the most important example being *Runtuwene et al.* (2018) – we do feel we have made significant scientific advances with our work.

First, and for the reasons you mention, we agree that the performance of primer design software is very difficult to rigorously evaluate. However, we do not believe that there are a large number of programs developed for multiplex PCR primer design that are (1) capable of designing for distributed targets (rather than tiling small genomes); (2) open-source and (3) actively maintained. The most well-known software in this space, PrimalScheme, currently only allows for tiling of smaller viral genomes. Consistent with this, the three major amplicon panels designed for *P. falciparum* to date (Broad Institute: <https://onlinelibrary.wiley.com/doi/10.1111/1755-0998.13622>, UCSF: <https://academic.oup.com/jid/article/225/7/1227/5897005>, MalariaGEN: <https://www.malariagen.net/resources/amplicon-sequencing-toolkit/p-falciparum-amplicontoolkit-protocols>) all used commercial and/or non-publicly available software. Anecdotally, we find many researchers invest considerable time trying to design multiplex panels in an *ad hoc* fashion because they cannot find suitable software. Finally, note that Reviewer #1 highlights in his first major comment that MULTIPLY meets a need of many researchers; a statement that supports the idea that not much software equivalent to MULTIPLY is available.

Second, we feel the multiplex PCR panels we developed with the software represent a major advance. Technically, there is no precedent for 3–4kbp multiplex PCRs containing 8 and 16 targets, in the context of the (A+T)-rich *P. falciparum* genome, and covering the complete coding sequences of major drug-resistance, diagnostic, and vaccine targets. Although *Runtuwene et al.* (2018) designed 1-3kbp amplicons for a similar set of 9 genes, each was produced by an individual PCR which made the assay impractical to scale to many samples. Simultaneously, we have worked to substantially reduce the costs of the library preparation. These two features – a multiplex PCR covering policy-relevant targets and a cost-effective protocol – will enable important malaria genetic data to be generated with the MinION on a scale that was formerly not possible.

Third, on top of performing careful validation of sequencing coverage, the influence of parasitemia, and SNP calling accuracy – we have demonstrated that our assay enables some novel and interesting analyses. In Figure 5 we have characterised the within-sample diversity of infections using the *msh2* gene, in part with a novel implementation of the Needleman-Wunsch algorithm that uses the Q-scores of ONT reads. In response to the review, we have also explored our ability to detect *hrp2/3* deletions causing false-negative RDT results, in the process developing a novel statistical method for inference (Figure 6).

Overall, we feel the novel and open-source primer design software, the two long-read multiplex PCR panels, cost-efficient protocol for MinION, careful validation on mock and field samples with the latest chemistry/algorithms, and development of novel statistical approaches for characterising within-sample diversity and calling *hrp2/3* deletions together represent a significant scientific advance.

Major points:

1. Since the initial application of the nanopore sequencer for the PCR-amplicon sequencing of Ebola virus (Quick et al Nature 2016), there has been a long history of the sequenced-based genome typing analysis for various pathogens on the nanopore platform. Several papers have also reported the analysis of drug resistant genes in malaria parasite (Runtuwene et al, Sci Rep 2018). In my opinion, in spite of those methodological developments, the “on-site” sequencing has not yet become the “standard” approach. There are several reasons, but the largest one may be that the sample preparation (DNA extraction, WGA and PCR amplification; above all the library preparation for the sequencing) is not easy to be completed in a field condition in most of the cases. For the research purpose, practically, the samples are transferred to a “central” lab at some stage. But, once they are transferred to the laboratory, the short read sequencing (of shotgun amplicons) is the easier and more cost-effective method. On the medical practice base, I wonder if the real-time PCR should be the first choice, at the first place, once the key mutations have been identified. I’m not convinced the authors have successfully addressed these issues in this particular paper.

First, we would like to clarify that we do not believe direct “on-site” sequencing, i.e. at the point of sample collection, is a useful standard approach in countries where malaria is endemic (although it is possible and may be useful in limited scenarios, for example in remote outbreaks). There are, however, many small- and medium-sized laboratories around Africa with a strong interest in pathogen genomics (for both research and surveillance) but that lack the facilities or capital to purchase and maintain expensive short-read sequencing platforms. In these cases, the MinION is preferable and is indeed becoming the standard approach for multiple pathogens (e.g. SARS-CoV-2, see <https://www.science.org/doi/10.1126/science.abq5358>). For *P. falciparum*, the only available method for the last five years has been Runtuwene et al. (2018) yet, as we mention above, due to costs and the number of individual PCRs required, it was not suitable for large scale surveillance. Our protocol has addressed these issues and we are productively deploying it in a small container lab at the National Malaria Elimination Centre in Zambia. Therefore we are confident there is a context for nanopore-based sequencing of *P. falciparum* – namely small- to medium-sized laboratories – where demand is high, and the sample / library preparation is not prohibitive.

While the use of short-read sequencing platforms in central laboratories will have higher throughput, we disagree that the sample or library preparation is any easier. Moreover, capital costs are hundreds of times that for nanopore, Illumina’s maintenance contracts in Africa are extremely expensive, and to achieve a better per-sample cost vs nanopore often requires running very large sample numbers which is not always appropriate. Anecdotally, we have visited several “central” labs in Africa which have Illumina machines that are not being used for a combination of these reasons. Real-time PCR approaches are indeed easier to implement than both short- or long-read sequencing, but have the disadvantage of being limited to focusing on individual SNPs. There are certainly cases where this is sufficient, but also many cases where it is not – in our example there are at least 30 SNPs of interest; and for *Kelch13* in particular new phenotypically important SNPs are constantly emerging.

In response to your concerns (and similar concerns Reviewer #2), we further emphasise these points in the last paragraph of the discussion.

2. Long read nature of the nanopore sequencing may provide a different potential from the other short read platforms. However, the amplicon size presented here is not particularly long, and setting long targets would rapidly decrease the success rate of the amplification. Also, having a haploid genome at the blood stage, the advantage in using the long read sequencing for the malaria genotyping may be limited rather than other pathogens having diploid genomes or having a high rate of co-infections.

It is true 3-4kbp is not particularly long compared to some of the reads generated today. However, compared to short-read assays that are prevalent for *P. falciparum*, 3 – 4kbp is sufficient to enable several new and important analyses, including: (i) spanning of the entire coding sequence of critical drug-resistance targets; (ii) read-backed phasing in co-infections which will enable better phenotype prediction; and (iii) investigation of repeat-

containing or very low-complexity genes, such as the antigen domain of malaria vaccine target *csp*. None of these analyses are feasible using existing short-read *P.f.* panels, whose amplicons are ~10-20X smaller.

At the same time, 3 – 4kbp strikes a good balance by avoiding being too long which, as you point out, would cause issues with consistent amplification from dried-blood spot (DBS) samples.

Finally, although it is true that *P. falciparum* is haploid within the host, the rate of co-infection is substantial, exceeding 50% in much of Africa (see: <https://elifesciences.org/articles/40845>).

3. Rather than the PCR-based selection of the target regions, making use of the “adaptive sampling” function of nanopore sequencing should be, at least, easier. Have the authors tried this approach?

We have not tried adaptive sampling *in lieu* of PCR-based selection for *P. falciparum*, although we are excited by the technology, and agree it could make the preparation of samples for sequencing much easier. However, for our use case – cost-efficient sequencing of a small set of *P.f.* genes – it is unlikely it would work well.

To appreciate this, the key reference is *Martin et al. (2022) Genome Biology* “Nanopore adaptive sampling: a tool for enrichment of low abundance species in metagenomic samples”. -In this paper, the authors derive and validate a formula for the expected amount of enrichment achieved by adaptive sampling, as a function of five parameters:

Parameter	Description	Value
R	Average read length	Experiment specific (bp)
y	Fraction of molecules in the original sample that the experimenter wants to enrich	Experiment specific (fraction)
S	Speed of sequencing	400 bp per second
D	Time it takes to make an accept/reject decision	1 second
C	Time it takes to capture a read once a pore is empty	0.5 seconds

The enrichment, α , is defined as the fold-increase in the fraction of reads that are from target regions when adaptive sampling is used, compared to when it is not used:

$$\alpha = \frac{\frac{y}{R} + \frac{1-y}{R}}{\frac{y}{R} + (1-y)\frac{D}{R} + \frac{C}{R}}$$

From this formula the authors show that the maximum amount of enrichment theoretically possible is $1/y$. This makes logical sense: if the target regions comprise 5% of the original reads, the most you can enrich by is 20x, as this would mean you *only* sequenced target regions.

However, from the same formula you can actually derive a second bound on the maximum enrichment possible that is much tighter for cases where the fraction to enrich is small. Notice from above that as long as $\frac{y}{R} > \frac{C}{R}$ (which is true when the read length is >400bp), the maximum enrichment will always be achieved as $\frac{D}{R}$ approaches zero, as this minimises the denominator. At $\frac{D}{R} = 0$, the formula reduces to:

$$\alpha_{\max} = \frac{\frac{y}{R} + \frac{1-y}{R}}{\frac{y}{R} + \frac{C}{R}}$$

From this formula, you can see the critical factor dictating maximum enrichment is how long it takes to accept/reject a read vs how long it takes to sequence one; as well as the read capture time. Intuitively, the

quicker an accept/reject decision is relative to the time it takes to sequence a read, the more time is saved by adaptive sampling, and the greater the enrichment. Below is a plot as a function of read length:

Unfortunately, you can see with the current technology these enrichment levels are still reasonably modest. Even if we were able to extract genomic DNA (gDNA) from dried-blood spots of mean 20kbp length and achieved the theoretical maximum enrichment we would still only expect ~34X enrichment. If our infection was 10,000 parasites per uL the *P.f.* gDNA would comprise ~1% of the total gDNA and, here, our target regions represent only ~0.1% of that *P.f.* gDNA (30kbp or so from 23Mbp). Therefore, without adaptive sampling we would expect to sequence target region reads at a rate of about 1 in 100,000; with adaptive sampling this would be 34 in 100,000. Note this is the absolute theoretical maximum, in practice (e.g. with error in accept/reject decisions) it would likely be much

worse.

Ultimately, it would be almost impossible to achieve sufficient coverage levels, or else only very few samples could be sequenced. Consistent with this, there is a recent preprint where adaptive sampling was applied to enrich for whole *P.f.* genomes (<https://www.biorxiv.org/content/10.1101/2022.11.29.518068v1>). This is a better use case as the fraction to enrich is closer to optimal, but still the authors managed only to multiplex 4 samples per flow cell. Given the current flow cell costs, this would not be a cost-effective or scalable method.

In summary, at present adaptive sampling is not a viable replacement for PCR-based enrichment. However, as the technology continues to improve – in particular as the accept/reject decision time gets smaller – it may become a viable alternative in the future.

Minor points:

4. With the advent of the new flow cell (R10.4 and kit 14), I think it may be timely for additional analysis regarding Figure 4b, to see whether the new version is appropriate to handle homopolymer regions. For example, it has been known that MinION performance in AT-rich and homopolymer (as in the introns of pfCRT) regions are very erroneous, but its performance in the CDS is reliable.

We have repeated this analysis and updated Fig 4 with the R10.4.1 flow cell and kit 14 chemistry.

5. Eventual success rates of the diagnosis (on the absence or presence of the drug resistant mutations) should be presented in a more explicit manner, depending on the varying parasitemia and condition of the samples.

It is difficult to comprehensively assess the success rate of diagnosis because of the number of factors that influence this (parasitemia, sample set quality, complexity of infection, sequencing performance) is very large. As an alternative to this, we have focussed on what we believe is the critical variable influencing resistance calling rate: sequencing coverage. Our belief is that the impact of the other factors (parasitemia, quality, &c) is largely through their influence on coverage.

Miscellaneous points:

6. Please mention that DBS is the abbreviation of dry blood spot in the beginning of the manuscript.

This has been done, thank you.

7. The multiply software seems to require VCF file. If we are to create primers of unknown sequences, then VCF file may not be available. How do handle it in this situation? Or is VCF file is optional?

The VCF file is optional. If the user has a VCF file (or a BED file) representing SNPs, indels, or just sites to avoid placing primers, it can be included. But if not, this step will be skipped and not included in the final multiplex selection decision.

8. On page 7, it is written than 57.5% of total sequencing were on-target. But Figure 2a depicts 57.4%.

Thank you for pointing this out. We have corrected the discrepancy.

9. On page 8, section "Effect of parasitemia on sequencing performance", the authors evaluated the metrics for four sequencing experiments described before. Which four experiments did the authors refer to?

We have updated the text in this section to make the experiments included more clear to readers (as well as include additional experiments).

In the original draft, we were refer to the experiment described at the beginning of the previous section: "**Characterising sequencing efficiency and coverage across mock and field samples**". In particular, here:

"We characterised the coverage generated by our protocol by running experiments with both the NOMADS8 and NOMADS16 panels on two different sample sets. The first set included 24 mock samples, created *in vitro* from standard laboratory or cultured strains of *P. falciparum* malaria (Methods, Supplementary Table 2). The second was a set of 28 DBS assessed as *P. falciparum* positive by RDT, and collected from a clinical setting in Kaoma, Zambia (Methods)."

10. Also, in the same section, the authors mentioned that they did not perform any sample normalization, but y-axis of Figure 3 top panel depicts normalized sample throughput.

By "sample input normalisation", we meant we did not normalise the amount of each sample that was input into the protocol in the lab; for example, some protocols normalise the input volumes by parasitemia or DNA mass after extraction / PCR.

The per sample coverage normalisation in Fig 3 normalises with the total amount of coverage generated from the flow cell on which the sample was sequenced (i.e. the experiment or sequencing run). This is because our interest was looking at how parasitemia influenced coverage, but different flow cells have differing active pores which compounds this comparison across experiments.

We see how this is confusing and have tried to clarify the language in the text.

11. On page 10, the authors mentioned that 150 in-silico replicates, but the sample size for this experiment is 30. Please clarify.

This section has been updated with a new sequencing experiment using R10.4.1 flow cells, and in the process we have tried to communicate the replicate numbers more clearly. The key point is that the sample size in Fig 4b (now n=20) refers to the number of replicates *at each particular read depth*. As a result, each element in the Fig 4b is a *mean* F1-score, averaged across the n=20 replicates performed at that depth. Since we have n=20 *in silico* replicates for each read depth, and we explored downsampling to ten different depths (10x, 20x, ... 100x) we have 200 *in silico* replicates total.

We also have added a sentence in the figure caption to help clarify this point..

12. Does Clair3 software create a consensus sequence of the reads before calling the SNPs?

No it does not, the approach uses a machine learning algorithm to call variants assuming a diploid sample.

Reviewers' Comments:

Reviewer #1:

Remarks to the Author:

The authors have addressed most of my comments well, and overall the manuscript has greatly improved.

My remaining comment refers to the off-target binding site identification that multiply performs. Given the human genome is 100-fold larger than the Plasmodium genome, and in most cases more copies of the human genome are present, when reading the manuscript I would assume that 'off-target' also refers to the human genome. On page 4, the text should be expanded as follows:

"[...] and identifies potential off-target binding sites using blastn against the P. falciparum reference genome. Multiply does not check potential binding sites in the human genome, or the genome of other blood-borne pathogens".

Reviewer #2:

Remarks to the Author:

I appreciate that you have taken time to review suggestions made including where they required generation of new data and I believe the manuscript is stronger as a result of it.

Reviewer #3:

Remarks to the Author:

First of all, I appreciate the substantial efforts of the authors for this revision. In fact, due to the careful validations and a substantial number of extensive analyses, the manuscript has been very much improved. Honestly, I still have a remaining concern to what extent the proposed approach should be scaled to meet the practical needs in Africa. However, I understand that part should wait for the future work of the authors. I sincerely hope this paper should be the first step for paving the way towards that goal. Anecdotally, I myself have also seen the Illumina sequencers left un-used because of several practical reasons, although the initial cost for this instrument has been enormous. I think how to make use of the nanopore sequencer, for which the initial cost is much lower, should be the key for more prevailed sequence-based diagnosis in developing countries. I foresee a number of a substantial barrier, but I believe in the continuous efforts of the authors. I look forward to seeing the first fruit to bear in Zambia.

Our response is in blue.

REVIEWERS' COMMENTS

Reviewer #1 (Remarks to the Author):

The authors have addressed most of my comments well, and overall the manuscript has greatly improved.

Thank you for taking the time to carefully review our manuscript.

My remaining comment refers to the off-target binding site identification that multiply performs. Given the human genome is 100-fold larger than the Plasmodium genome, and in most cases more copies of the human genome are present, when reading the manuscript I would assume that 'off-target' also refers to the human genome. On page 4, the text should be expanded as follows:

"[...] and identifies potential off-target binding sites using blastn against the *P. falciparum* reference genome. Multiply does not check potential binding sites in the human genome, or the genome of other blood-borne pathogens".

We have made this change to the manuscript text. Following from your concerns, we plan to add a "contaminating genomes" feature to MULTIPLY in the near future. This will allow users to list, in the design file, other genomes that might be in the sample – so that primers are also blasted against these genomes and any discovered off-target binding sites are added to the final cost function.

Reviewer #2 (Remarks to the Author):

I appreciate that you have taken time to review suggestions made including where they required generation of new data and I believe the manuscript is stronger as a result of it.

Thank you for your review and your protocol improvement suggestions.

Reviewer #3 (Remarks to the Author):

First of all, I appreciate the substantial efforts of the authors for this revision. In fact, due to the careful validations and a substantial number of extensive analyses, the manuscript has been very much improved. Honestly, I still have a remaining concern to what extent the proposed approach should be scaled to meet the practical needs in Africa. However, I understand that part should wait for the future work of the authors. I sincerely hope this paper should be the first step for paving the way towards that goal. Anecdotally, I myself have also seen the Illumina sequencers left un-used because of several practical reasons, although the initial cost for this instrument has been enormous. I think how to make use of the nanopore sequencer, for which the initial cost is much lower, should be the key for more prevailed sequence-based diagnosis in developing countries. I foresee a number of a substantial barrier, but I believe in the continuous efforts of the authors. I look forward to seeing the first fruit to bear in Zambia.

Thank you for your review and encouraging reflections. We agree that there are still barriers to scaling nanopore-based approaches for *P.f. malaria* in Africa, but we also have confidence that with the continued effort of ourselves and the wider community, these can be overcome.

Reviewer #3 (Remarks on code availability):

Even though I'm not fully qualified to evaluate the algorithmic/theoretical aspects, the code should be practically. And for the practical use, it is not that difficult to follow the procedure.